# Characterization of Lung and Oral Microbiomes in Lung Cancer Patients Using Culturomics and 16S rRNA Gene Sequencing

Yifan Sun,[a] Yuejiao Liu,[a] Jianjie Li,[b] Yafang Tan,[a] Tongtong An,[b] Minglei Zhuo,[b] Zhiyuan Pan,[a] Menglei Ma,[b] Bo Jia,[b] Hongwei Zhang,[b] Ziping Wang,[b] Ruifu Yang,[a] (ID) Yujing Bi[a]

[a]State Key Laboratory of Pathogen and Biosecurity, Beijing Institute of Microbiology and Epidemiology, Beijing, China
[b]Department of Thoracic Oncology, Peking University Cancer Hospital, Beijing, China

Yifan Sun and Yuejiao Liu contributed equally to this work. Author order was determined by the corresponding author after negotiation.

**ABSTRACT** Recently, microbiota dysbiosis in lung cancer has attracted immense attention. Studies on lung microbes are mostly based on sequencing, which has left the potentially functional bacteria with extremely low abundance uncovered. In this study, we characterized and compared the lung and oral cavity microbiotas using culturomics and 16S rRNA gene sequencing. Of the 198 bacteria identified at the species level from bronchoalveolar lavage fluid (BALF) samples, *Firmicutes* was predominant (39.90%). Twenty bacterial species isolated from BALF samples were present in at least half of the patients and were also highly abundant in oral samples. Of all isolated strains, *Streptococcus* and *Veillonella* were highly dominant. The abundance of *Prevotella* and *Veillonella* decreased from the oral cavity to the lung, whereas that of *Pseudomonas* increased. Linear discriminant analysis effect size demonstrated that *Prevotella* was more abundant in the healthy samples than in the cancerous ones, which is in accordance with the isolation of *Prevotella oralis* only from the healthy group using culturomics. Moreover, *Gemella sanguinis* and *Streptococcus intermedius* were isolated only from the non-small-cell lung cancer (NSCLC) group, and 16S rRNA gene sequencing showed that they were higher in the NSCLC than in the small-cell lung cancer group. Furthermore, while *Bacillus* and *Castellaniella* were enriched in lung adenocarcinoma, *Brucella* was enriched in lung squamous cell carcinoma. Overall, alterations were observed in the microbial community of patients with lung cancer, whose diversity might be site and pathology dependent. Using culturomics and 16S rRNA gene amplicon sequencing, this study has provided insights into pulmonary and oral microbiota alterations in patients with lung cancer.

**IMPORTANCE** The relationship between lung microbiota and cancer has been explored based on DNA sequencing; however, culture-dependent approaches are indispensable for further studies on the lung microbiota. In this study, we applied a comprehensive approach combining culturomics and 16S rRNA gene amplicon sequencing to detect members of the microbiotas in saliva and BALF samples from patients with unilateral lobar masses. We found alterations in the microbial community of patients with lung cancer, whose diversity might be site and pathology dependent. These features may be potential bacterial biomarkers and new targets for lung cancer diagnosis and treatment. In addition, a lung and oral microbial biobank from lung cancer patients was established, which represents a useful resource for studies of host-microbe interactions.

**KEYWORDS** microbiota, lung cancer, BALF, oral bacteria, culturomics, 16S rRNA gene, DNA sequencing, lung infection, oral microbiota

Address correspondence to Ziping Wang, wangzp2007@126.com, Ruifu Yang, 13801034560@163.com, or Yujing Bi, byj7801@sina.com.

The authors declare no conflict of interest.

Lung cancer, the most common type of cancer worldwide, is closely associated with chronic inflammation (1). Inflammation caused by microbial infections might contribute to cancer development and progression (2). Polymorphic microbiomes have recently been

added as one of the four new "hallmarks" of cancer (3). Increasing evidence shows that the lung microbiome is involved in cancer pathogenesis.

Healthy lungs, conventionally considered sterile, are now known to harbor a diverse microbiota (4). To date, several culture-independent analyses have reported the association between microbial population diversity and lung cancer (5, 6). Lung tissue samples from lung cancer patients demonstrated an increase in $\alpha$ diversity and the phylum *Bacteroidetes* compared with lung tissue samples from emphysema patients (7). A study found that the genus *Streptococcus* was more abundant in the bronchoalveolar lavage fluid (BALF) samples from patients with lung cancer than in those of healthy controls (8). Few studies have shown the association between lung bacteria and the histological subtypes of lung cancers; the genera *Veillonella*, *Megasphaera*, *Enterobacter*, *Morganella*, and *Klebsiella* were significantly higher in lung adenocarcinoma (ADC) than in lung squamous cell carcinoma (SCC) (9, 10). As the oral cavity is the entry point to the respiratory tract, its microbiome may contribute to lung cancer (11). Tsay et al. reported that lower airway dysbiosis induced by microaspiration of oral commensals promotes an interleukin-17-driven inflammatory phenotype, which exacerbates lung tumorigenesis (12). *Sphingomonas* and *Blastomonas* were shown to be more abundant in the oral microbiomes of lung cancer patients (13). However, the possible variations in the oral and lung microbiotas of lung cancer patients and the differences in the microbial diversity of their saliva and BALF samples remain unclear.

While numerous DNA sequencing-based investigations have explored the relationship between the lung microbiota and cancer, they have several inherent drawbacks, such as depth bias and a high detection threshold (14). Therefore, culture-dependent approaches are indispensable for further studies of the lung microbiota. Culturomics, in which multiple culture conditions are combined for rapid identification, provides new perspectives on host-bacterium relationships (15). However, the use of culturomics for the culture and identification of bacteria in BALF has rarely been reported.

In this study, we applied a comprehensive approach combining culturomics and 16S rRNA gene amplicon sequencing to the saliva and BALF samples from 25 patients with unilateral lobar masses. To the best of our knowledge, this is the first study reporting the bacterial diversity and richness in the oral and BALF microbiotas of patients with lung cancer, using culturomics and 16S rRNA gene sequencing.

## RESULTS

**Clinical information.** We recruited 25 patients with unilateral lobar masses from Beijing Peking University Cancer Hospital between January 2021 and May 2022, of which 23 were newly diagnosed with lung cancer via histological confirmation but 2 were not. There were 16 non-small-cell lung cancer (NSCLC) patients, including 8 with ADC, 7 with SCC, and one with nonspecified NSCLC, and 7 small-cell lung cancer (SCLC) patients. None had previously received any anticancer therapy, radiation therapy, or antibiotic treatment. Of the 23 patients with lung cancer, 14 were men and 9 were women. Furthermore, 18 were smokers. There were 17 and 6 patients with and without distant metastasis, respectively (Table 1).

**Characteristics of bacteria isolated from the lung and oral cavity via culturomics.** Figure 1 shows the culturomics workflow. Briefly, total 45 samples were collected from 15 lung cancer patients, and four culture conditions were tested for each sample. We obtained 12,379 colonies from the BALF samples (from both cancerous [C] and healthy [H] sites), and 198 bacteria were identified at the species level using matrix-assisted laser desorption ionization–time of flight (MALDI-TOF) or 16S rRNA gene sequencing. The identified species belonged to six phyla, including *Firmicutes* (39.90%), *Proteobacteria* (27.78%), *Actinobacteria* (19.19%), *Bacteroidetes* (9.60%), *Fusobacteria* (2.02%), and *Synergistetes* (1.52%) (Fig. 2A and B). Comparison with the previously established repertoire of microorganisms isolated from the human gut, urine, vagina, and oral/respiratory tract revealed that approximately 1/4 of the species isolated in this study had been isolated previously from these four sites (47/198 [23.7%]) (Fig. 2C).

We obtained 5,671 colonies from the oral samples and identified 156 bacterial species

**TABLE 1** Clinical characteristics of patients

| Variable | Value for patient type | | |
|---|---|---|---|
| | NSCLC | SCLC | Noncancer |
| No. | 16 | 7 | 2 |
| Age [mean (SD)] | 65 (7.0) | 67 (12.8) | 69 (8.2) |
| | | | |
| Gender [no. (%)] | | | |
| Male | 9 (56) | 5 (72) | 2 |
| Female | 7 (44) | 2 (14) | 0 |
| | | | |
| Smoking [no. (%)] | | | |
| Current or former smoker | 12 (75) | 6 (86) | 1 |
| Nonsmoker | 6 (25) | 1 (14) | 1 |
| | | | |
| Pathological diagnosis [no. (%)] | | | |
| Adenocarcinoma | 8 (50) | | |
| Squamous cell carcinoma | 7 (44) | | |
| Unidentified | 1 (6) | | |
| | | | |
| Distant metastasis [no. (%)][a] | | | |
| M0 | 4 (25) | 2 (29) | |
| M1 | 12 (75) | 5 (71) | |

[a]M0, without metastasis; M1, with metastasis.

(see Fig. S1 in the supplemental material). At the phylum level, the bacterial diversity of the oral sample was similar to that of BALF, consisting predominantly of *Firmicutes* and *Proteobacteria* (>66%). In addition to the previously known bacteria, we isolated 15 potential new species (Table S1).

**Comparison of microbiotas in lung and oral cavity.** We compared the microbiota composition between the C and H lung sites and the oral cavity (O) using 16S rRNA gene amplicon sequencing. There was a noticeable difference in the microbiota composition between the lung and oral cavity. In the lung samples, we observed a moderate difference between the C and H samples at the phylum level (Fig. 3A). The three dominant phyla were *Proteobacteria*, *Firmicutes*, and *Bacteroidetes*. At the genus level, *Pseudomonas* (*Proteobacteria*), *Streptococcus* (*Firmicutes*), *Veillonella* (*Firmicutes*), and *Prevotella_7* (*Bacteroidetes*) were the most common ones in the BALF samples. Contrastingly, *Prevotella_7* (*Bacteroidetes*), *Neisseria* (*Proteobacteria*), *Streptococcus* (*Firmicutes*), *Veillonella* (*Firmicutes*), and *Haemophilus* (*Proteobacteria*) were the most common ones in the oral samples (Fig. S2A). We did not observe any significant difference in the richness or diversity of the microbial community ($\alpha$ diversity) between the BALF samples from C and H sites, as measured using the Shannon index ($P = 0.527$) and Chao1 index ($P = 0.428$). Moreover, there was no difference in the overall microbiota ($\beta$ diversity) between the C and H groups, as measured using the Bray-Curtis distances ($P = 0.39$) (Fig. S3). Nevertheless, the oral sample was significantly different from both the C and H lung samples in $\alpha$ and $\beta$ diversity (Fig. 3B and C). Metastats analysis at the genus level further revealed unique anatomy-related microbial

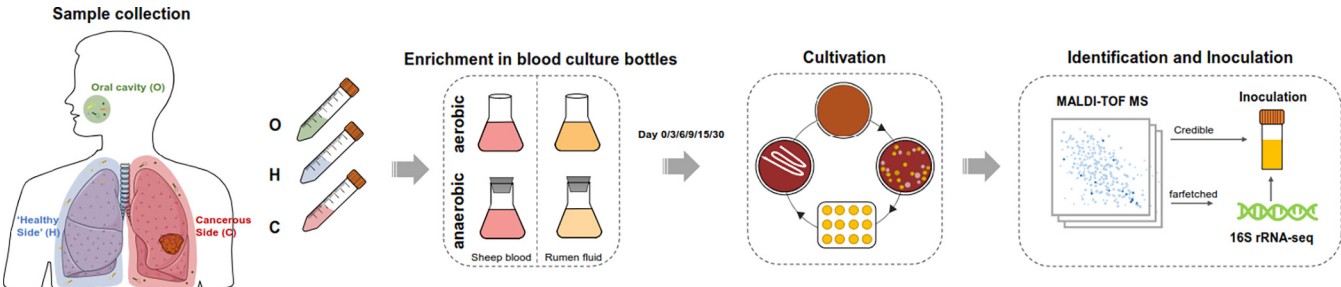

**FIG 1** Summary of culturomics methods and workflow.

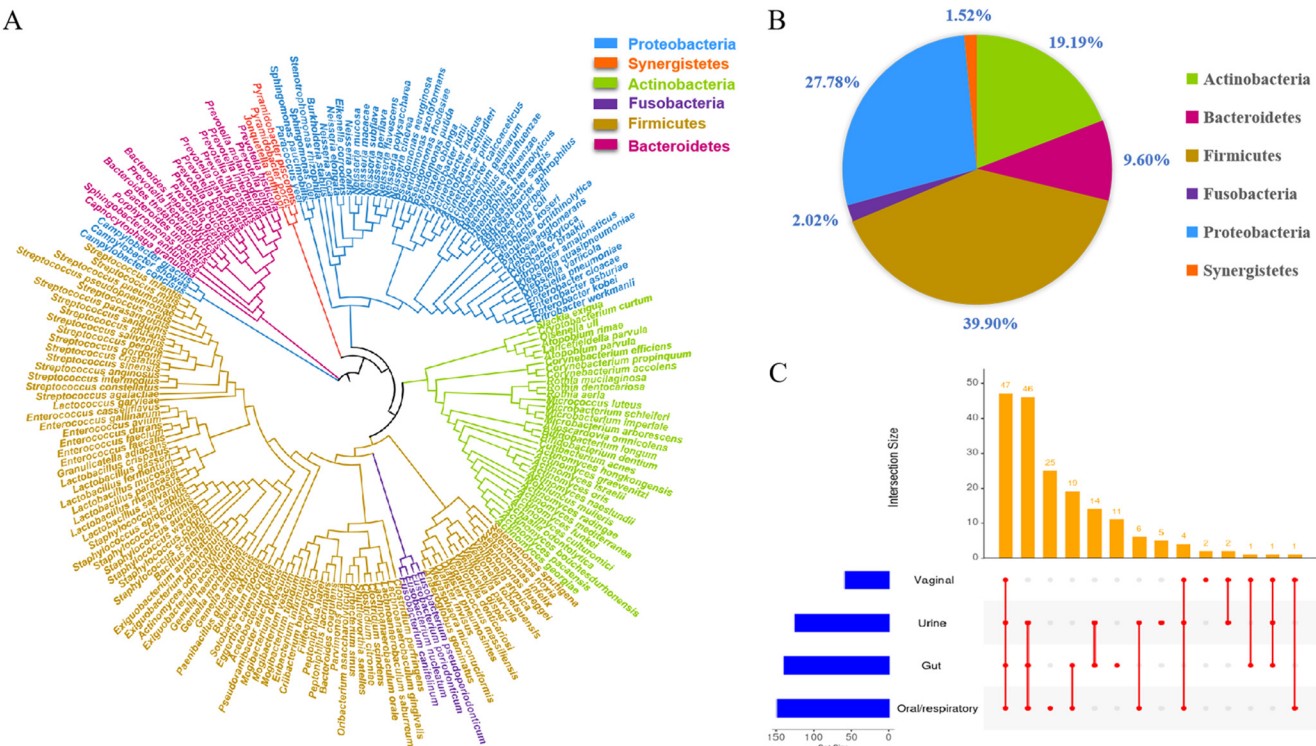

**FIG 2** Bacteria identified from the BALF samples. (A) Phylogenetic tree and (B) proportion of 198 bacterial species isolated from the BALF samples listed according to phylum. (C) UpSet plot showing the shared cultured species among human oral/respiratory, gut, urine, and vagina samples.

features, such as higher abundance of *Prevotella* in oral samples (Fig. 3D). *Pseudomonas* was the only genus concentrated in BALF samples (Fig. 3D; Fig. S2B).

Furthermore, we compared the bacteria isolated via culturomics among different groups. Unlike the sequencing data, there was no obvious difference in the bacterial proportion at the phylum level among C, H, and O samples, with *Firmicutes* being dominant in all three sites (Fig. 3E). More than half of the species (89) were isolated in all three sites, and 34, 32, and 39 unique species were isolated from the C, H, and O groups, respectively (Fig. 3F). In addition, we analyzed the prevalence of bacteria in the lungs and oral cavity. Based on the culturomics results, we defined the strains isolated from over 50% of the patients as prevalent strains. Twenty prevalent strains were identified from the BALF samples, which belonged to 12 genera (Table 2). *Streptococcus* was the major genus, and *Streptococcus oralis*, *Veillonella atypica*, *Parvimonas micra*, and *Actinomyces odontolyticus* were found in almost all BALF samples. These 20 species were also cultured at a high frequency from oral samples, which indicated that the pulmonary microbiota might originate from the oral cavity.

**Differences in the microbiota composition between cancerous and healthy lungs.** To compare the relative contributions of different taxa, we used linear discriminant analysis (LDA) effect size (LEfSe) to detect taxa with differential abundances between the two groups We identified 14 different taxa at various levels with significantly different abundances between the two groups, of which four were differentially abundant at the genus level. *Prevotella* and *Prevotella* 7 were more abundant in the H group, whereas *Carnobacterium* and *Brucella* were more abundant in the C group (Fig. 4A).

Moreover, we compared the differences between the H and C groups of bacteria obtained via culturomics. We drew a heat map of the proportion of each bacterium in the total sample. *Streptococcus oralis*, *Veillonella atypica*, and *Parvimonas micra* were cultured at a high frequency without significant differences. A significantly higher frequency of *Prevotella oralis* was found in the H group ($P = 0.019$) (Fig. 4B), which agreed with the sequencing results.

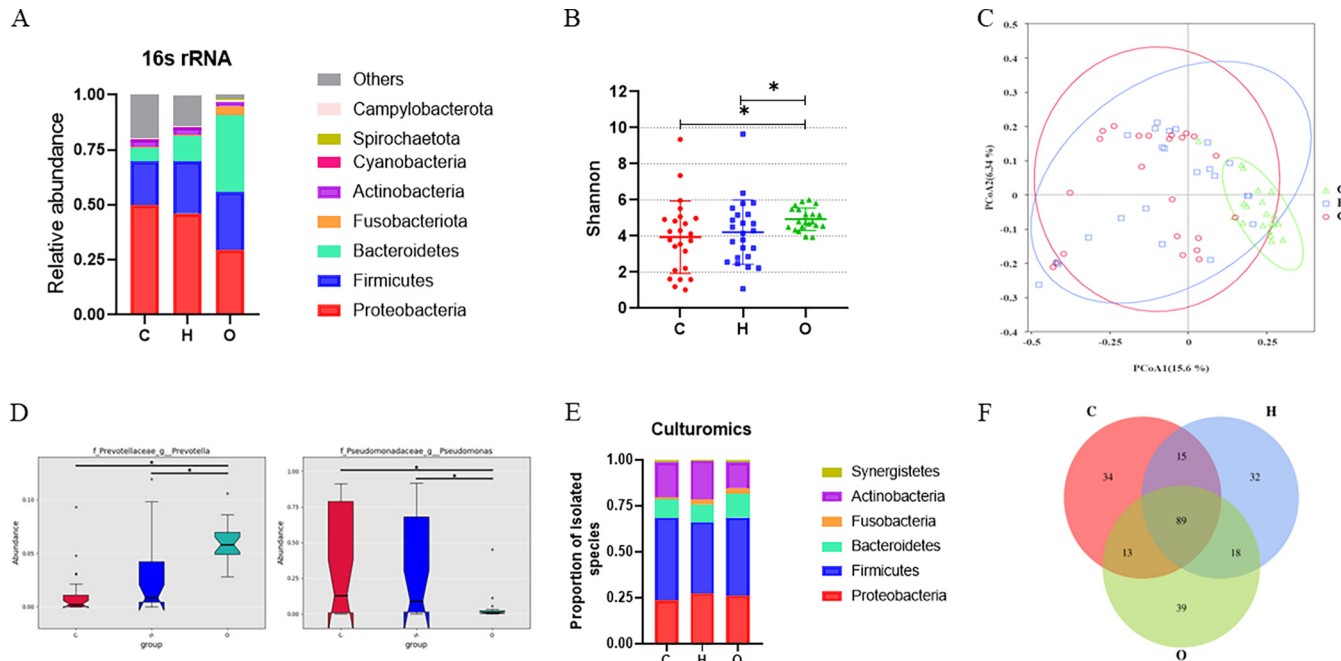

**FIG 3** Variations in microbial composition in different anatomical sites. (A) Taxonomic composition at the phylum level in BALF and oral samples based on 16S rRNA gene amplicon sequencing (top 30). (B) Shannon diversity index of BALF and oral samples. *P* values were calculated using the Wilcoxon test. *, *P* < 0.05. (C) PCoA of samples from three different anatomical site in patients with lung cancer. ANOSIM was performed to statistically evaluate significant difference. (D) Notched box plots illustrating the differences of the significantly 2 genus relative abundances in three different anatomy sites. *, *P* < 0.05. (E) Proportions of bacterial species isolated from C, H, and O samples listed according to phylum. (F) Venn diagram of the culturable bacterial species from the three anatomical sites. C, H, and O, samples from the cancerous site and contralateral healthy controls from lungs and the oral site, respectively, of patients with lung cancer.

**Different pathological strains reshape different pulmonary microbiomes.** We examined the differences in the lung microbiotas of NSCLC and SCLC groups with comparable basic information, including age, sex, smoking status, and distant metastasis (Table S2). Bacteria with an abundance of >4% were considered dominant. *Pseudomonas* (32%), *Streptococcus* (5%), and *Bacillus* (4%) were the dominant genera in the 32 NSCLC lung samples, while in SCLC samples, the dominant genera were *Pseudomonas* (31%), *Veillonella* (7%), and *Prevotella* 7 (6%) (Fig. 5A). The α diversity of NSCLC was not significantly different from that of SCLC, as measured using the Shannon index (*P* = 0.931) and Chao1 index (*P* = 0.72), or β diversity, as measured using the Bray-Curtis distances (*P* = 0.489) (Fig. S4A and B). The culturomics study revealed that 77 and 33 species were isolated only from NSCLC and SCLC lung samples, respectively (Fig. 5B). Only bacterial species isolated from at least half of the patients in the group were compared. Later, we analyzed the bacteria isolated from only one group with an isolation rate of ≥50%. *Gemella sanguinis*, *Pseudoramibacter alactolyticus*, *Bifidobacterium dentium*, and *Streptococcus intermedius* were the four species isolated with a relatively high frequency only from the NSCLC group. *Prevotella pallens* was the only species isolated at a relatively high frequency from the SCLC lung samples. Subsequently, we compared the relative abundances of these five bacteria at the genus level using 16S rRNA gene amplicon sequencing. Although there was no significant difference between the two groups, their abundance tended to be higher in the site from which they were isolated (Fig. 5C).

In the subtype analysis of patients with NSCLC, *Pseudomonas* (26%) and *Bacillus* (8%) were dominant in the 16 ADC samples. In contrast, in the 14 SCC samples, the dominant genera were *Pseudomonas* (38%) and *Streptococcus* (8%) (Fig. 5D). Shannon index (*P* = 0.27) and Chao1 index (*P* = 0.402) were not significantly different between the ADC and SCC groups. However, β diversity showed that the microbiota constitution of the ADC lung samples was significantly different from that of the SCC samples (*P* = 0.021) (Fig. S4C and D). In the LEfSe analysis of ADC and SCC, 11 taxa displayed contrasting correlations between the NSCLC subtypes. A differential abundance analysis at the genus level between ADC and SCC

**TABLE 2** Summary of the proportion of the top 20 bacteria in BALF samples and the related proportions in the oral cavity

| Genus | Species | Frequency of detection (%) in sample type[a] | | |
|---|---|---|---|---|
| | | C | H | O |
| Streptococcus | Streptococcus salivarius | 60.00 | 60.00 | 73.33 |
| | Streptococcus pseudopneumoniae | 53.33 | 40.00 | 33.33 |
| | Streptococcus parasanguinis | 93.33 | 86.67 | 100.00 |
| | Streptococcus oralis | 93.33 | 93.33 | 100.00 |
| | Streptococcus mitis | 93.33 | 86.67 | 100.00 |
| | Streptococcus gordonii | 53.33 | 53.33 | 80.00 |
| | Streptococcus constellatus | 66.67 | 73.33 | 93.33 |
| | Streptococcus anginosus | 66.67 | 66.67 | 100.00 |
| Veillonella | Veillonella parvula | 60.00 | 60.00 | 73.33 |
| | Veillonella atypica | 66.67 | 80.00 | 66.67 |
| Solobacterium | Solobacterium moorei | 66.67 | 60.00 | 73.33 |
| Slackia | Slackia exigua | 60.00 | 73.33 | 60.00 |
| Parvimonas | Parvimonas micra | 80.00 | 80.00 | 93.33 |
| Mogibacterium | Mogibacterium diversum | 53.33 | 46.67 | 53.33 |
| Granulicatella | Granulicatella adiacens | 66.67 | 53.33 | 46.67 |
| Gemella | Gemella morbillorum | 66.67 | 53.33 | 66.67 |
| Fusobacterium | Fusobacterium nucleatum | 40.00 | 66.67 | 80.00 |
| Dialister | Dialister invisus | 73.33 | 53.33 | 73.33 |
| Anaeroglobus | Anaeroglobus geminatus | 53.33 | 60.00 | 73.33 |
| Actinomyces | Actinomyces odontolyticus | 80.00 | 80.00 | 66.67 |

[a]C, H, and O, samples from the cancerous lung sites and contralateral healthy control and the oral site, respectively, of patients with lung cancer.

showed enrichment of *Bacillus* and *Castellaniella* in patients with ADC, whereas those with SCC had a higher abundance of *Brucella* (Fig. 5E).

## DISCUSSION

The relationship between the microbiota and cancer is being extensively investigated (16). Accumulating evidence indicates that the gut microbiota contributes to carcinogenesis and response to immunotherapy (16, 17). While many studies have focused on analyzing the influence of the gut microbiota, its composition differs substantially from that of the lung microbiota (18, 19). Although microaspiration commonly occurs in healthy subjects, it is more frequent in those with chronic inflammatory airway diseases (20, 21). The entry of more oral microbes into the lungs is associated with the increased release of lung proinflammatory cytokines and a proinflammatory phenotype characterized by elevated Th-17 lymphocytes (22, 23). Lung microbiota dysbiosis might modulate the risk of malignancy at multiple levels, including chronic inflammation and activation of oncogenes (24, 25). Although the lung microbiome has been implicated in lung cancer in several specific ways, its exact role in carcinogenesis has not yet been elucidated. One of the possible reasons for this is that current studies are mostly based on sequencing levels, which results in a lack of strain materials for further study. Here, we utilized both culturomics and 16S rRNA gene amplicon sequencing to evaluate and compare the structure and diversity characteristics of oral and pulmonary microbiotas associated with lung cancer.

Culturomics can identify bacteria at strain level, and successfully isolating living bacteria is crucial for carrying out subsequent experimental work (14, 26). A previous study summarizing a list of microbes isolated from the human lower respiratory tract showed that the lung microbiome was dominated by the phyla *Pseudomonadota*, *Firmicutes*, *Bacteroidota*, and *Actinomycetota*, which was consistent with our study (27). Here, we cultured 198 identified bacterial species from human BALF and 156 from oral samples from lung cancer patients, including 15 potentially new taxa. A separate article describing the novel bacteria, new.4 and new.10, belonging to a novel genus, which we named "*Curtanaerobium*" (28). The present study enabled us to expand the human respiratory and oral repertoire. We

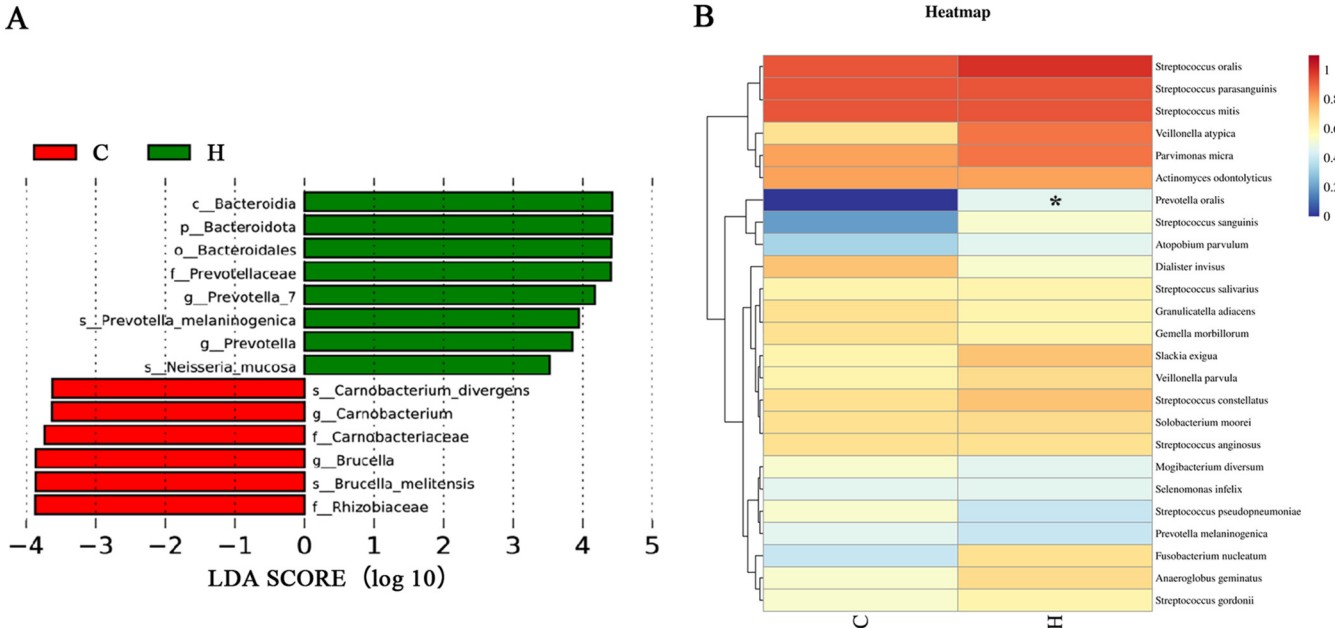

**FIG 4** Differentially abundant taxonomy between cancer and paired healthy lung. (A) LEfSe was used to identify the bacterial microbiota that significantly differed between the cancerous and healthy lungs. Only taxa meeting a significant LDA threshold value of >3.5 and having a $P$ value of <0.05 are shown. (B) Heat map analysis of lung microbiotas at C and H sites of patients with lung cancer based on culturomics ($n$ = 15). Each row represents an individual species with the higher isolated rate in the two groups. Top 25 bacterial species were recovered from either group. Transition from blue to red represents the frequency of culturomics recovery from 0 to 1. Statistical comparisons were made using Fisher's exact test with FDR correction. *, $P < 0.05$.

also found 20 prevalent strains in both BALF and oral samples from patients with lung cancer, which indicates that the pulmonary microbiota might originate from the oral cavity. Culturomics might reduce the number of these unclassed or no-rank operational taxonomic units (OTUs) by increasing the number of pure-cultured microbial species. In this study, *Pseudoramibacter alactolyticus* was recovered with a relatively high frequency only from the NSCLC group but was not detected in 16S rRNA gene amplicon sequencing.

*Parvimonas micra* is an opportunistic pathogen that is frequently associated with several human infections and also causes purulent infections in multiple organs (29, 30). *P. micra* is highly abundant in patients with colorectal cancer and promotes colorectal tumorigenesis by inducing colonocyte proliferation and altering the Th17 immune response (31, 32). Our previous results showed that *P. micra* promoted colorectal cancer progression by upregulating miR-218-5p expression and ultimately activating the Ras/ERK/c-Fos signaling pathway (33). However, its correlation with lung cancer has not yet been reported. In this study, *P. micra* was cultured at a high frequency in patients with lung cancer, which provides valuable information for further studies.

We analyzed the microbiota composition at the genus level via taxonomic analysis, and the results showed that the representative flora differed based on the sampling site. We found that *Streptococcus*, *Veillonella*, and *Prevotella* were enriched in the oral samples, whereas *Pseudomonas* was enriched in the BALF samples. *Streptococcus* and *Veillonella* which were considered oral commensals were reported to increase in lower airways of lung cancer patients (34). *Pseudomonas* species, commonly found in the respiratory tract, are involved in the pathogenesis of lung diseases, such as chronic obstructive pulmonary disease and cystic fibrosis (35, 36). A study demonstrated that transplanted lungs are susceptible to the growth of multiple *Pseudomonas* species (37). *Pseudomonas aeruginosa* possesses numerous virulence factors that attack the respiratory epithelial cells, such as the type III secretion system (38). *P. aeruginosa* possesses pili and flagella that are necessary for motility and respiratory infection, as they enable attachment to the respiratory epithelium via respiratory mucins and the glycolipid asialo-GM1 (39). A study reported that *P. aeruginosa* was significantly more abundant in NSCLC patients with brain metastasis than in those without metastasis (40). Likewise, most of the patients in our study had distant

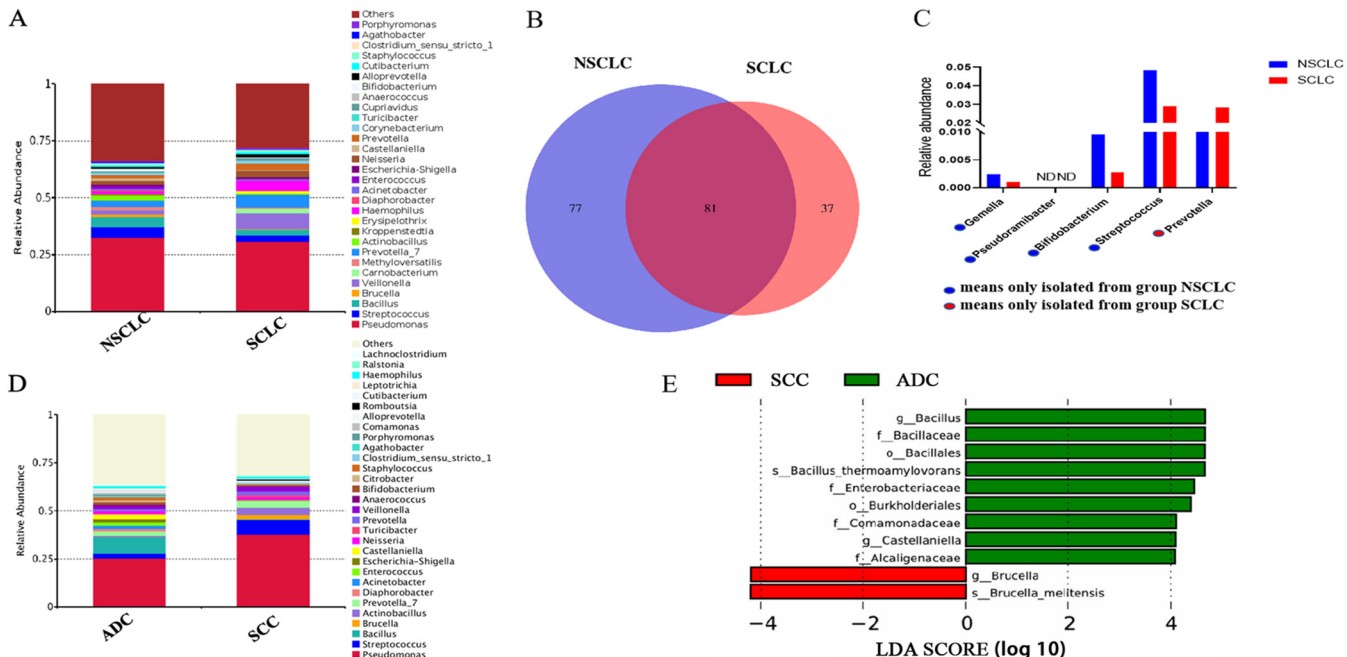

**FIG 5** Characterization of lung microbiotas in different lung cancer subtypes. (A) Taxonomic composition at the genus level in NSCLC and SCLC lung samples. (B) Venn diagram of the culturable bacterial species isolated only from NSCLC and only from SCLC lung samples. (C) Differences in the relative abundance of the genera corresponding to the species isolated from NSCLC and SCLC samples. Differential microbial taxa were identified using a paired *t* test, and the *P* values were adjusted for multiple comparison using the FDR. (D) Taxonomic composition at genus level in ADC and SCC lung samples. (E) LEfSe was used to identify the bacterial microbiotas that significantly differed between NSCLC and SCLC. Only taxa meeting a significant LDA threshold value of >4 and having a *P* value of <0.05 are shown.

metastasis, and *Pseudomonas* was the most prevalent genus in the BALF samples. These findings, including ours, suggest that *Pseudomonas* frequently colonizes the respiratory tracts of patients with lung cancer and might promote distant metastasis.

Based on 16S rRNA gene sequencing, we found that *Brucella melitensis* was more enriched in the cancerous lung than in the healthy lung. The genus *Brucella* includes several significant human and veterinary pathogens, which predominantly maintain an intracellular lifestyle in their mammalian hosts. These pathogens exhibit numerous pathogen-associated molecular patterns that can be recognized by innate receptors in airway epithelial cells (41). Their type IV secretion system secretes effector molecules into the host cell cytoplasm, which direct the intracellular trafficking of the brucellae and modulate host immune responses (42). Thus, we speculated that *Brucella melitensis* is a possible lung cancer biomarker.

*Gemella sanguinis*, *Pseudoramibacter alactolyticus*, and *Streptococcus intermedius* are involved in human inflammatory diseases, such as endocarditis (43), periodontal infections (44), and brain abscesses (45). Germ-free mice or mice treated with antibiotics had significantly lower incidence of lung cancer than specific-pathogen-free mice (46, 47). Our results showed that these three species were isolated only from the lung microbiotas of patients with NSCLC and not from those with SCLC. This observation was consistent with the results of 16S rRNA gene amplicon sequencing, which indicated that specific bacteria might mediate the development of NSCLC by inducing chronic inflammation.

When bacteria were categorized based on the lung cancer subtypes, *Bacillus* and *Castellaniella* were significantly enriched in the BALF samples from patients with ADC, whereas *Brucella* was enriched in the SCC group. Another study reported the enrichment of *Acinetobacter* in BALF samples from patients with ADC, whereas *Bacillus* and *Brucella* showed no significant differences between the ADC and SCC groups (10). These conflicting associations of microbiomes with different pathological subtypes could be attributed to the significant variations across individuals and sampling methods.

The lung microbiome is similar to other microbiomes in other parts of the human body, and there are significant individual differences. Although we screened some differential microorganisms via culture or sequencing, the sample size was small, and more clinical samples are needed for verification. The resolution of 16S rRNA gene sequencing is limited, since only bacterial identification at the genus level was relatively accurate, limiting our comparison between the two methods to genus information and ignoring the species information from the culturomics method. Metagenomic sequencing combined with culturomics should be preferable for the parallel comparison at the species level. Moreover, the causal relationship between the screened microorganisms and the occurrence and development of lung cancer remains uncertain, and follow-up studies are also required.

In conclusion, we examined the pneumonic and oral microbiotas of lung cancer patients using culturomics and 16S rRNA gene sequencing. We found alterations in the microbial community of patients with lung cancer, whose diversity might be site and pathology dependent. Using culturomics, we found that *Streptococcus* and *Veillonella* were highly dominant in both pneumonic and oral samples from patients with lung cancer, which suggests the possible deleterious effects of airway microbial dysbiosis originating from the oral cavity. We showed that *Prevotella oralis* was isolated only from the H group and that *Gemella sanguinis* was isolated only from the NSCLC group, which was consistent with the findings of 16S rRNA gene sequencing. This study provides basic data on the microbiota diversity in pneumonic and oral samples from patients with lung cancer. These organisms might serve as potential bacterial biomarkers and new targets for lung cancer diagnosis and treatment, and the causative relationships can be explored using the isolated strains.

## MATERIALS AND METHODS

**Patient recruitment and sample collection.** This study was approved by the Institutional Review Board of Peking University School of Oncology, China, and informed consent was obtained from all participants. The BALF samples were collected as previously described (48). The saliva samples were collected before the patients underwent bronchoscopy examination. All the participants were instructed to not eat and drink for 1 h prior to saliva sample collection. Twenty-five patients with unilateral lobar masses who consented to undergo bronchoscopic examination at Peking University Cancer Hospital were enrolled. All patients underwent transbronchoendoscopy to avoid contaminating the upper respiratory tract or oral microbiota, and paired BALF samples (one each from the cancerous site [C] and the contralateral healthy lung [H]) were collected before the operation. None of the participants had recently suffered from oral disease. Before the bronchoscopy, oral samples from 21 patients were collected. The sample was divided into two parts: one part (including 45 samples from 15 lung cancer patients) was used for culturing bacteria, and the other part (all of the 71 samples) was used for 16S rRNA gene amplicon sequencing. Fresh samples were collected in a sterile tube, placed on ice, and transported to the laboratory within 1 h. Culturomics was performed immediately after sample collection in the laboratory, and the aliquots were stored at −80°C before high-throughput sequencing.

**Culturomics. (i) The process of culturomics.** Culturomics is a high-throughput method that multiplies culture conditions to detect high bacterial diversity and pure bacterial cultures. This analysis involved preculturing under different conditions: aerobically with 5% sheep blood, aerobically with 5% rumen fluid, anaerobically with 5% sheep blood, and anaerobically with 5% rumen fluid. Sample dilution, strain isolation, and identification were performed as previously described (49). On days 1, 3, 6, 9, 15, and 30, the enriched culture samples were extracted from the bottles using a syringe. Subsequently, 100-μL doubling dilutions were spread onto Columbia agar supplemented with 5% sheep blood at 37°C under aerobic conditions for 24 h or anaerobic conditions for 72 h. Colonies were picked and identified using MALDI-TOF mass spectrometry (MS) systems (Autof MS1000). Colonies that could not be identified using the MALDI-TOF MS database V1.1.12 (score < 9) were subjected to 16S rRNA gene sequencing with the primers 27F (5′-AGAGTTTGATCMTGGCTCAG-3′) and 1492R (5′-GGTTACCTTGTTACGACTT-3′). The sequencing results were analyzed using the NCBI BLAST algorithm for homologous sequence searches with type strains. If the 16S rRNA gene sequence is <98.65% similar to the closest type strain, the isolate might be a new species (50).

**(ii) Classification of cultivated species.** We classified all isolates into four categories: oral/respiratory (this study and reference 27), gut (51), urine (52, 53), and vagina (54, 55). We also performed literature searches on PubMed to compare our results with the published data to confirm the classification.

**DNA extraction, 16S rRNA gene amplification, and sequencing.** DNA was extracted from each sample using a HiPure bacterial DNA kit (Mageon, China) according to the manufacturer's instructions. The V3-V4 region of the 16S rRNA gene was amplified using specific primers (341F, 5′-CCTAYGGGR BGCASCAG-3′; 806R, 5′-GGACTACNNGGGTATCTAAT-3′). Sequencing libraries were generated using a TruSeq DNA PCR-free sample preparation kit (Illumina, USA) following the manufacturer's recommendations, and index codes were added. The library quality was assessed with the Qubit 2.0 fluorometer

(Thermo Scientific) and Agilent Bioanalyzer 2100 system. The sequencing was performed with on an Illumina NovaSeq 6000 sequencing platform (Illumina, San Diego, CA, USA), and 250-bp paired-end reads were generated.

**Sequence data analysis.** Raw reads were filtered to remove adaptors, low-quality reads, and ambiguous bases. Clean data were extracted from the raw data using USEARCH 8.0. The OTUs were classified based on 97% similarity after chimeric sequences were removed using UPARSE (version 7.0.1001 [http://drive5.com/uparse/]), and the representative sequence from each OTU cluster was obtained. The $\alpha$ diversity was assessed using the nonparametric Shannon and Simpson indices; the $\beta$ diversity was calculated using Bray-Curtis distances in QIIME and visualized by principal-coordinate analysis (PCOA). LEfSe was used to detect taxa with differential abundance among groups. Metastats analysis was employed to detect the differences in the microbiota composition at the genus level.

**Statistical analyses.** Statistical analyses were performed using the R software (v3.4.10) and SPSS 20.0. Independent $t$ and chi-square tests were used to analyze the demographic and clinical data. Fisher's exact test with false-discovery-rate (FDR) correction was used to compare frequency of isolated species between groups. Sample diversity metrics were assessed based on the nonparametric Shannon index and Chao1 index. PCoA plots were generated to visualize the separation of samples based on pairwise distances, and analysis of similarity (ANOSIM) was performed to evaluate the statistically significant difference in PCoA. $P$ values of $<0.05$ were considered statistically significant.

**Ethics approval and consent to participate.** All procedures performed in studies involving human participants or animals were approved by the Institutional Review Board of the Peking University School of Oncology (no. 2018KT89). All patients gave their written consent before inclusion in the study.

**Data availability.** All the data generated or analyzed in this study are included in this article (or its supplemental material). The read sequences obtained from Illumina NovaSeq were submitted to the NCBI Sequence Read Archive (SRA) under accession number PRJNA904049 (BioProject ID).

## SUPPLEMENTAL MATERIAL

Supplemental material is available online only.

**SUPPLEMENTAL FILE 1**, PDF file, 0.6 MB.

## ACKNOWLEDGMENTS

We gratefully acknowledge all of the participants and the medical staff who helped collect the samples in this study.

We declare no competing interests.

This research was supported by the National Key Research and Development Program of China (2021YFC2301000), and the National Natural Science Foundation of China (31970863 and 81790632).

Y.B. designed research and project outline. R.Y. and Z.W. directed the research. Y.S., J.L., Y.L., T.A., M.Z., M.M., B.J., and H.W. performed isolation, deposition, and identification. Y.T. and Z.P. performed genome analysis. Y.S. and Y.B. drafted the manuscript. All authors read and approved the final manuscript.

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
