## [Reviewer comments · Microbiology Spectrum]

Microbiology Spectrum

Characterization of lung and oral microbiomes in lung cancer patients using culturomics and 16S rDNA sequencing

yifan sun, Yujing Bi, hongwei zhang, yuejiao liu, jianjie li, Tongtong An, bo jia, minglei zhuo, menglei ma, Ziping Wang, yafang tan, Ruifu Yang, and zhiyuan pan

Corresponding Authors: Ziping Wang, Peking University Cancer Hospital, and Ruifu Yang and Yujing Bi, State Key Laboratory of Pathogen and Biosecurity

Review Timeline:

Submission Date:	January 19, 2023
Editorial Decision:	February 11, 2023
Revision Received:	March 14, 2023
Accepted:	April 3, 2023

Editor: Diyan Li

Reviewer(s): Disclosure of reviewer identity is with reference to reviewer comments included in decision letter(s). The following individuals involved in review of your submission have agreed to reveal their identity: Raunak Kumar Das (Reviewer #5); Hassan Zafar (Reviewer #10); Pierre Le Bars (Reviewer #12); Pablo Alejandro Millones Gómez (Reviewer #20)

Transaction Report:

DOI: <https://doi.org/10.1128/spectrum.00314-23>

February 11, 2023

Dr. Yujing Bi
Beijing Institute of Microbiology and Epidemiology
Beijing
China

Re: Spectrum00314-23 (Characterization of lung and oral microbiome in lung cancer patients by culturomics and 16S rDNA sequencing)

Dear Dr. Yujing Bi:

1. p should be represented properly in the manuscript as it is a statistical symbol.
2. Notch box plot should be used instead of box whisker plot.
3. The no of clinical samples may be increased.
4. The authors should also comment on the keystone species and species diversity of each microbiome and their changes.

Link Not Available

Sincerely,

Diyang Li

Journals Department
Reviewer comments:

Reviewer #10 (Comments for the Author):

The manuscript titled, "Characterization of lung and oral microbiome in lung cancer patients by culturomics and 16S rDNA sequencing" by Sun et al provides valuable insight about the bacterial populations in the lung and oral bacteriome in lung cancer patients. Overall, I firmly believe the study provides important information which can be translated to future animal models, and also pave the way for studies that may use the identified bacterial species as biomarkers. The methods are also well explained and the bioinformatic tools were utilized well in the study. However, I have some major and minor comments for the

authors to consider. I believe the manuscript will be much improved if the authors incorporate my suggestions.

Major comments

- The authors use the word "cultruomics" throughout the Text, but I believe the word is "culturomics". Is it a typo throughout the manuscript? I think the authors must address this point, because the word is a part of the manuscript title.
- I firmly believe the manuscript requires extensive English editing. I have come across numerous grammatical errors and typos. I have tried to highlight most of them in the minor comments, but still, I have the feeling that the text requires further editing.
- In the discussion section, I think the authors should add some information about the bacterial species that were identified in the cancer patients. For instance, why *Pseudomonas* is such a potential threat in lung cancer, is it due to certain pathogenicity factors that are encoded by its genome? Similarly, I think adding some information about the other identified species will provide more rigor to the Discussion section.

Minor comments

- Line 25, mention the full name of BALF not just the abbreviation
- Line 26, "Twenty" not 20.
- Line 27, "BALF" instead of "BLAF".
- Line 31, mention the full name of "LEfSE".
- Line 33, "Prevotella Oralís" not "Prevotella orails"
- Line 39, "found" not "founded".
- Line 41, add "this study" before "provides"
- Line 56-58, should be rephrased.
- Line 67, add "an" instead of "the"
- Line 68 has some spacing issues.
- Line 81 again "BLAF".
- Line 87, add a full stop after 1
- Line 93, "Twenty five" instead of "25"
- Line 93, I think "from patients" is a mistake.
- Line 95, the word should be "transbronchoendoscopy"
- Line 98-99, replace ", total 21 patients' oral samples were collected" with "oral samples from 21 patients were collected".
- Line 113, "100 µL" not "100ul"
- Line 116, mention the full name of "MALDI-TOF MS".
- Line 118 has a strange spacing kindly check.
- Line 141 "raw" instead of "Raw".
- Line 143 add "were" after "sequences"
- Line 171, replace "were not" with "did not".
- Line 176, check if the word "cultruoims" is correct?
- Line 182, I think the exact values should be mentioned that are given in the Figure 2A.
- Line 188, you mention that 156 bacterial species were identified, but there is not mention of this in the Supplementary information.
- Line 197-198, rewrite the sentence. It has grammatical errors.
- Line 204-205, its hard to understand the meaning, kindly rephrase.
- Line 220, add "of" after "prevalence"
- Line 240, "results" instead of "result".
- Line 241, "prevalence" not "prevalent"
- Line 242, "Different pathological types shaped different pulmonary microbiomes" the name should be changed to "Different pathological strains reshape different pulmonary microbiomes"
- Line 318, again "BLAF"
- Line 325, add "prevalent" before "genus".
- Line 331, there is a double space before "Germ-free".
- Line 336, "bacteria" not "bacterial".
- Line 363, "procedures" not "Procedures".

Reviewer #12 (Comments for the Author):

The objective of this research aimed to analyze the lung and oral microbiome of patients with lung cancer.

There is a lack of information concerning the patients (oral pathologies?). The conclusions must be accompanied by several reservations concerning the size of the sample, the natural biodiversity of the microbiome between patients. Comparing the microbiomes (oral and pulmonary) of the same patient from a cancerous lung area with another non-cancerous area of the same patient cannot be 100% reliable.

In the study, a significant segregation was found at the level of the genus, but without having sought the equivalent level at the species. The taxonomic levels of the genus are not the relevant biological measurement units for some authors (BIK, ME et al. Bacterial diversity in the oral cavity of 10 healthy individuals) *International society for microbial ecology*. (2010). However in the mouth, the level of ecological interest chosen to label is the genus rather than species. There also reservations concerning the

conclusions of the article are to be made.

See the other remarks in the attached PDF. Corrections on the form and the bottom are necessary. Several reservations may call into question the validity of the result.

Add Bibliography.

Ramirez-Labrada, AG et al. The influence of microbiota on lung carcinogenesis immunity and immunotherapy . Trends Cancer 2020. 6, 86-97.

Yagi, K.; Huffnagle, G.B.; Lukacs, N.W.; Asai, N. The Lung Microbiome during Health and Disease. Int. J. Mol. Sci. 2021, 22, 10872. <https://doi.org/10.3390/ijms221910872>

Dickson, R.P.; Erb-Downward, J.R.; Martinez, F.J.; Huffnagle, G.B. The Microbiome and the Respiratory Tract. Ann. Rev. Physiol. 2016, 78, 481-504. [CrossRef]

Pizzo F. et al. Role of the microbiota in lung cancer insights on prevention and treatment. Int Mol Sci. 2022 23. (11) 6138.Q. et al Alterations of fecal bacterial communities in patients with lung cancer. Am J Transl 2018. 10 3171 -3185.

Zhang WQ. Et al 2018. Alterations of fecal bacterial communities in patients with lung cancer. Georgiou K et al . Gut microbiota and lung cancer. Where Do we stand? Int J mol Sci 2021.

Goubet AG. Et al The impact of the intestinal microbiota in therapeutic responses against cancer. Comptes Rendus Biol 2018. 341 284-289.

Segal LN et al. enrichment of the lung microbiome with oral taxa is associated with lung inflammation of Th17 phenotype. Nat Microbiol 2016.

Mortensen MS. The developing hypopharyngeal microbiota in early life. Microbiome 2016.

Das S. et al . A prevalent and culturable microbiota links ecological balance to clinical stability of the human lung after transplantation. Nat Commun 2021.

Reviewer #20 (Comments for the Author):

This is an interesting study aimed at characterizing and associating the oral and pulmonary microbiome of lung cancer patients. However, the study has not considered that the oral microbiome depends mainly on the oral condition. It has been widely demonstrated that high bacterial richness in the salivary microbiota is significantly associated with poor oral health, as indicated by decayed teeth, periodontitis, and poor oral hygiene. Therefore, it becomes crucial to understand oral microbial diversity and how it fluctuates under conditions of disease/disturbance. Advances in metagenomics and next-generation sequencing techniques generate rapid sequences and provide extensive information on the microorganisms inhabiting a niche. Therefore, the information retrieved can be used to develop microbiome-based biomarkers for use in the early diagnosis of oral and associated diseases. However, a homogenization of the oral clinical conditions of the samples must be considered for the results to be robust.

Reviewer #5 (Comments for the Author):

1. p should be represented properly in the manuscript as it is a statistical symbol.

2. Notch box plot should be used instead of box whisker plot.
3. The no of clinical samples may be increased.
4. The authors should also comment on the keystone species and species diversity of each microbiome and their changes.

Staff Comments:

Preparing Revision Guidelines

Please return the manuscript within 60 days; if you cannot complete the modification within this time period, please contact me. If you do not wish to modify the manuscript and prefer to submit it to another journal, please notify me of your decision immediately so that the manuscript may be formally withdrawn from consideration by Microbiology Spectrum.

The manuscript titled, “Characterization of lung and oral microbiome in lung cancer patients by culturomics and 16S rDNA sequencing” by Sun et al provides valuable insight about the bacterial populations in the lung and oral bacteriome in lung cancer patients. Overall, I firmly believe the study provides important information which can be translated to future animal models, and also pave the way for studies that may be use the identified bacterial species as biomarkers. The methods are also well explained and the bioinformatic tools were utilized well in the study. However, I have some major and minor comments for the authors to consider. I believe the manuscript will be much improved if the authors incorporate my suggestions.

Major comments

- The authors use the word “cultruomics” throughout the Text, but I believe the word is “culturomics”. Is it a typo throughout the manuscript? I think the authors must address this point, because the word is a part of the manuscript title.
- I firmly believe the manuscript requires extensive English editing. I have come across numerous grammatical errors and typos. I have tried to highlight most of them in the minor comments, but still, I have the feeling that the text requires further editing.
- In the discussion section, I think the authors should add some information about the bacterial species that were identified in the cancer patients. For instance, why *Pseudomonas* is such a potential threat in lung cancer, is it due to certain pathogenicity factors that are encoded by its genome? Similarly, I think adding some information about the other identified species will provide more rigor to the Discussion section.

Minor comments

- Line 25, mention the full name of BALF not just the abbreviation
- Line 26, “Twenty” not 20.
- Line 27, “BALF” instead of “BLAF”.
- Line 31, mention the full name of “LEfSE”.
- Line 33, “*Prevotella Oralis*” not “*Prevotella orails*”
- Line 39, “found” not “founded”.
- Line 41, add “this study” before “provides”
- Line 56-58, should be rephrased.
- Line 67, add “an” instead of “the”
- Line 68 has some spacing issues.
- Line 81 again “BLAF”.
- Line 87, add a full stop after 1
- Line 93, “Twenty five” instead of “25”
- Line 93, I think “from patients” is a mistake.
- Line 95, the word should be “transbronchoendoscopy”
- Line 98-99, replace “, total 21 patients’ oral samples were collected” with “oral samples from 21 patients were collected”.
- Line 113, “100 μ L” not “100ul”

- Line 116, mention the full name of “MALDI-TOF MS”.
- Line 118 has a strange spacing kindly check.
- Line 141 “raw” instead of “Raw”.
- Line 143 add “were” after “sequences”
- Line 171, replace “were not” with “did not”.
- Line 176, check if the word “cultruoims” is correct?
- Line 182, I think the exact values should be mentioned that are given in the Figure 2A.
- Line 188, you mention that 156 bacterial species were identified, but there is not mention of this in the Supplementary information.
- Line 197-198, rewrite the sentence. It has grammatical errors.
- Line 204-205, its hard to understand the meaning, kindly rephrase.
- Line 220, add “of” after “prevalence”
- Line 240, “results” instead of “result”.
- Line 241, “prevalence” not “prevalent”
- Line 242 , “Different pathological types shaped different pulmonary microbiomes” the name should be changed to “Different pathological strains reshape different pulmonary microbiomes”
- Line 318, again “BLAF”
- Line 325, add “prevalent” before “genus”.
- Line 331, there is a double space before “Germ-free”.
- Line 336, “bacteria” not “bacterial”.
- Line 363, “procedures” not “Procedures”.

**Characterization of lung and oral microbiome in lung cancer patients**
**by culturomics and 16S rDNA sequencing**

Yifan Sun^{1,#}, Yuejiao Liu^{1,#}, Jianjie Li², Yafang Tan¹, Tongtong An², Minglei Zhuo²,
Zhiyuan Pan¹, Menglei Ma², Bo Jia², Hongwei Zhang², Ziping Wang^{2,*}, Ruifu Yang^{1,*},
Yujing Bi^{1,*}

1. State Key Laboratory of Pathogen and Biosecurity, Beijing Institute of
Microbiology and Epidemiology, Beijing 100071, China;

2. Department of thoracic oncology, Peking University Cancer Hospital, Beijing
100142, China

#These authors contributed equally to this work.

* Corresponding authors: Ziping Wang (wangzp2007@126.com)

Ruifu Yang (13801034560@163.com)

Yujing Bi (byj7801@sina.com);

**Abstract:**

Microbiota dysbiosis in lung cancer has attracted more attention recently. However,
most relevant studies on lung microbes are based on sequencing, making the
potentially functional bacteria with extremely low abundance uncovered. Here, we
employed both culturomics and 16S rDNA sequencing to characterize and compare
microbiota in lung and oral cavity. From BALF samples, 198 bacteria at species level
were isolated and Firmicutes predominated (40%). 20 bacterial species isolated from
BLAF samples occurred in at least half of patient which were also present in a high
proportion of oral samples. *Streptococcus* and *Veillonella* were the highly dominant
group among all isolated strains. We showed that the abundance of genus *Prevotella*
and *Veillonella* displayed a decreased trend from oral site to lung, whereas
*Pseudomonas* increased. LEfSE analysis demonstrated that *Prevotella* increased in
“Healthy” side (H) than Cancerous side (C), which was consistent with the species
*Prevotella orails* only isolated from H group by culturomics. Moreover, *Gemella*
*sanguinis*, and *Streptococcus intermedius* were only isolated from non-small cell lung
cancer (NSCLC) group while the 16S rDNA amplicon sequencing also showed a
tendency that they are higher in the NSCLC than small cell lung cancer (SCLC)
group. Furthermore, *Bacillus* and *Castellaniella* were enriched in lung
adenocarcinoma (ADC), but *Brucella* in lung squamous cell carcinoma (SCC).
Overall, we founded that the microbial community changed in lung cancer patients,
which diversity might be site- and pathological-dependent. Combined with
culturomics and 16S rDNA amplicon sequencing provides deeper insights about the
pulmonary and oral microbiota changes of lung cancer patients.

**Keywords:** Microbiota, Lung cancer, BALF, Oral bacteria, Culturomics, 16S rDNA

Introduction

Lung cancer is the most common cancer worldwide, and is closely associated with
chronic inflammation [1]. Inflammation caused by microbial infection may contribute
to cancer development and progression [2]. Polymorphic microbiomes have been
recently added to one of the four new “Hallmarks” of cancer [3]. Evidence is
mounting that the lung microbiome may play a role in cancer pathogenesis.

Healthy lungs that were traditionally thought to be sterile are now known to harbor
a diverse microbiota [4]. In date, a substantial number of studies applying culture-
independent analysis have reported that microbial population diversity was associated
with lung cancer [5, 6]. Richness of lung microbiota was reduced in lung cancer
patients but the composition of the bacterial flora of patients in phylum Bacteroidetes
was significantly higher compared with control subjects [7, 8]. A study found that
genus *Streptococcus* was more abundant in BALF samples of lung cancer patients
than healthy controls [9]. Few studies showed links between lung bacteria and
histological subtypes of lung cancers; the genera *Veillonella*, *Megasphaera*,
*Enterobacter*, *Morganella*, and *Klebsiella* were significantly higher in lung
adenocarcinoma than lung squamous cell carcinoma [10, 11]. Oral cavity is the entry
point for the respiratory tract, and the oral microbiome may contribute to lung cancer
risk [12]. Tsay et al. reported that lower airway dysbiosis induced by microaspiration
of oral commensals affects lung tumorigenesis by promoting an IL-17-driven
inflammatory phenotype [13]. A study showed that the oral microbiome of bacterial
genera *Sphingomonas* and *Blastomonas* were relatively higher in lung cancer
patients [14]. However, the possible variations of oral and lung microbiota in lung
cancer patients and the difference in microbial diversity in samples from the saliva
and BALF has yet to be defined.

While numerous DNA sequencing-based investigations have been performed to
explore the relationship between the lung microbiota and cancer, it has several
inherent drawbacks, such as depth bias and a high detection threshold [15]. Culture-
dependent approaches are indispensable for further studies of lung microbial function.
Culturomics, which uses multiple culture conditions combined with rapid
identification, is developed to provide new perspectives on host-bacteria
relationships [16]. However, culturomics has rarely been reported for the culture and
identification of bacteria in BALF.

In this work, we propose to apply a comprehensive approach combining
culturomics and 16S rDNA amplicon sequencing to saliva and BLAF samples from
25 lung patients with unilateral lobar masses. We reported the bacteria diversity and
richness of oral and BALF microbiota from lung cancer patients and applied
culturomics for the first time to culture and identify them.

**Materials and methods**

**1 Patient recruitment and samples collection**

This study was approved by the Institutional Review Board of the Peking University
School of Oncology, China, and informed consent was obtained from all subjects.
BALF sample collection was performed as previously described [8]. Before the
patients were selected for bronchoscopy examination, saliva samples were collected.
All the participants were instructed to not eat and drink for 1 h prior to saliva sample
collection. 25 patients with unilateral lobar masses were enrolled from patients who
consented to bronchoscopy examination at Peking University Cancer Hospital. All
patients underwent transbronchoendoscopy, which avoided contamination of the
upper respiratory tract or oral microbiota, and paired BALF samples (one from the
cancerous side (C), the other from the contralateral healthy lung (H)) were collected
before the operation. Before the bronchoscopy, total 21 patient oral samples were
collected. The sample was divided into two parts: one part (including 45 samples from
15 lung cancer patients) was used for culturing bacteria, and the other part (all of the
71 samples) was used for 16S rDNA amplicon sequencing. The fresh samples were
collected in a sterile tube, placed on ice, and transported to the laboratory within 1 h.
Cultruomics was carried out on the day of collection in the laboratory immediately,
while aliquots were then stored at -80°C before high-throughput sequencing.

**2 Cultruomics**

**2.1 Process of cultruomics**

Cultruomics is a high-throughput method that multiplies culture conditions to detect
higher bacterial diversity and pure bacterial cultures. This cultruomics study was pre-
cultured under different conditions: aerobically supplemented with 5% sheep blood;
aerobically supplemented with 5% rumen fluid; anaerobically with 5% sheep blood;

anaerobically with 5% rumen fluid. Sample dilution, strain isolation, and
identification were performed as previously described [17]. On days 1, 3, 6, 9, 15, and
30, samples of enriched cultures were extracted from the bottles by syringe, and 100ul
doubling dilutions were spread onto Columbia agar supplemented with 5% sheep for
culture at 37°C aerobic conditions for 24 h or anaerobic conditions for 72 h. Colonies
were picked and identified using MALDI-TOF MS systems (Autof MS1000).
Colonies that were not identified by MALDI-TOF MS systems of database V1.1.12
(score < 9) were subjected to 16S rDNA gene sequencing with primers 27F (5'-
AGAGTTTGATCMTGGCTCAG-3') and 1492R (5'-GGTTACCTTGTTACGACTT-
3'). Sequencing results were analyzed by the NCEBLAS algorithm4 for
homologous sequence searches with type strains. If 16S rDNA is <98.65% similar to
the closest type strain, the isolate could be a new species [18].

**2.2 Classification of cultivated species**

Classify all isolates into four categories: oral/respiratory(this study and [19]),
gut [20], urine [21, 22], and vagina [23, 24]. We also conducted literature searches on
PubMed to compare against published papers and confirm the classification.
Sequences of the 16S rDNA of isolated strains were aligned using the CLUSTALX
program. A phylogenetic tree was constructed using neighbor-joining with MEGA
version X with 1000 replicate bootstrap values. The classical tree visualization is
supported online tool (<https://itol.embl.de>) [25].

**3 DNA Extraction, 16S rDNA Amplification and Sequencing**

DNA was extracted from each sample using Hipure Bacterial DNA kit (Mageon,
China) according on the manufacturer's recommendation. The V3-V4 region of 16S
rDNA gene was amplified using specific primers (341F: 5'-
CCTAYGGGRBGCASCAG-3'; 806R: 5'-GGACTACNNGGGTATCTAAT-3').
Amplicons were sequenced using the Miseq platform. Sequencing was performed on
an Illumina Novaseq6000 sequencing platform (Illumina, San Diego, CA, United
States), and 250 bp paired-end reads were generated.

**4 Sequence data analysis**

Raw reads were filtered to remove adaptors and low-quality and ambiguous bases,

clean data were extracted from Raw data using USEARCH 8.0. Operational
taxonomic units (OTUs) were classified based on 97% similarity after chimeric
sequences removed using UPARSE (version 7.0.1001 <http://drive5.com/uparse/>) and
the representative sequence from each OTU cluster was obtained. Alpha diversity was
assessed on the basis of the nonparametric Shannon index and Simpson index. The β
diversity was calculated using Bray-Curtis distances in QIIME and visualized by
principal coordinate analysis (PCoA). The linear discriminant analysis (LDA) effect
size (LEfSE) was used to detect taxa with differential abundance among groups. The
Metastats analysis was employed to detect the differences in microbiota composition
among groups at the genus levels.

**5 Statistical analysis**

Statistical analysis was performed by R software (v3.4.10) and SPSS 20.0. For
demographic and clinical data, independent t test and chi-square test were used.
Fischer Exact test with FDR correction were used for higher isolated species between
the two groups. Sample diversity metrics were assessed on the basis of the
nonparametric Shannon–Wiener diversity index and Chao1 diversity index. To
visualize separation of samples based on pairwise distances, principal coordinate
analysis (PCoA) plots were generated, and anosim was performed to test statistically
whether there is a significant difference in PCoA analysis. All statistical analyses
were performed using R software (Version 2.15.3). *P* value < 0.05 was considered
statistically significant.

**RESULTS**

**1、 Clinical information**

A total of 25 patients with unilateral lobar masses were recruited between January
2021 and May 2022 from Beijing Peking University Cancer Hospital. Among all the
patients, 23 patients were newly diagnosed with lung cancer by histological
confirmation, but 2 patients were not. There were 16 non-small cell lung cancer
(NSCLC) including 8 adenocarcinoma (ADC), 7 squamous cell carcinoma (SCC), and

one with non-specified NSCLC, and 7 small cell lung cancer (SCLC). The lung
cancer patients were not previously received any anticancer therapy nor were treated
with any antibiotics. In 23 lung cancer patients, there were 14 males and 9 females,
and totally 18 smokers. There were 6 cases without distant metastasis and 17 with
distant metastasis (Table 1).

**2、 Characteristics of bacteria isolated from lung and oral cavity by**

**cultruomics**

The cultruomics workflow is shown in Figure 1. Briefly, total 45 samples from 15
lung cancer patients were collected, and 4 culture conditions were tested for each
sample. From BALF samples (including both cancerous lung (C) and the contralateral
“healthy” lung (H)), a total of 12379 colonies were obtained, and 198 bacteria at
species level were identified by MALDI-TOF or 16S rDNA gene sequencing. The
identified species belonged to 6 phyla, including Firmicutes (40%), Proteobacteria
(28%), Actinobacteria (19%), Bacteroidetes (10%), Fusobacteria (2%), and
Synergistetes (2%) (Figure 2A and B). By comparing the previously established
repertoire of isolated microorganisms from the human gut, urine, vagina, and
oral/respiratory tract, about 1/4 of the isolated species in this study were previously
isolated from 4 multiple sites in the human body (47/198, 23.7%) (Figure 2C).

From oral samples, a total of 5671 colonies were observed, and 156 bacteria at species
level were identified (Fig. S1). At the phylum level, the bacterial diversity of the oral
sample was similar to BALF, Firmicutes and Proteobacteria were predominated (>
66%).

In addition to previously known bacteria, 15 potentially new species were isolated
from this cultruomics study. (Table S1)

**3、 Comparison of microbiota in lung and oral cavity**

We compared the microbiota composition between cancerous-side lung (C),
“healthy”-side lung (H), and oral cavity (O) by 16S rDNA amplicon sequencing.
There was obvious difference between lung and oral cavity, but C and H seemed
moderate difference at phylum level (Figure 3A). Three dominant phyla were
Proteobacteria, Firmicutes, and Bacteroidetes. At genus level, *Pseudomonas*

(Proteobacteria), *Streptococcus* (Firmicutes), *Veillonella* (Firmicutes), and
*Prevotella_7* (Bacteroidetes) were the most common in the BALF samples. In
contrast, *Prevotella_7* (Bacteroidetes), *Neisseria* (Proteobacteria), *Streptococcus*
(Firmicutes), *Veillonella* (Firmicutes), and *Haemophilus* (Proteobacteria) were the
most common in oral samples (Figure S2A). In the BALF group, cancer lung was not
significantly from “healthy” lung in richness and diversity of microbial community (α
-diversity), as measured by the Shannon ($P=0.527$) and Chao1 diversity index
($P=0.428$), or overall microbiota (β -diversity), as measured by Bray-Curtis distances
($P=0.390$) (Figure S3). Nevertheless, the oral sample was significantly different from
both the cancerous side and the “healthy” side of the lung in α and β -diversity (Figure
3B, 3C). Metastat analysis based on genus level further revealed unique anatomy-
related microbial features, such as more abundant *Prevotella* in oral samples (Figure
3D). *Pseudomonas* was the only genus concentrated in BALF samples (Figure 3D,
S2B).

We also compared the bacteria isolated by culturomics between different groups.
Different with sequence data, there was no obvious difference for bacterial proportion
at phylum level between cancerous-side (C), “healthy”-side (H), and oral cavity (O).
Firmicutes predominated in all three sites (Figure 3E). More than half of the species
(89) were isolated in all three sites, and there was 34, 32 and 39 special species
isolated in C, H and O group respectively (Figure 3F). Further, we analyzed the
prevalence bacteria in lung and oral cavity. For culturomics results, we defined the
strains isolated from more than 50% of patients as prevalent strains. For BALF
samples, 20 species were identified as prevalent strains, which belonged to 12 genus
(Table 2). *Streptococcus* was the major genus, and *Streptococcus oralis*, *Veillonella*
*atypica*, *Parvimonas micra*, and *Actinomyces odontolyticus* were found in nearly all
BALF samples. Similar, these 20 species were also cultured at a high frequency in
oral samples, which indicated the pulmonic microbiota maybe come from oral cavity.

**4、Difference in microbiota compositions between cancerous and** 228 **“healthy” lung**

To compare the relative contribution of different taxa, we used the LEfSE to detect
taxa with differential abundance among the two groups. A total of 14 different taxa at

various levels with significant abundances across two groups were identified, of
which four differentially abundant taxa at the genus level were noted. *Prevotella* and
*Prevotella_7* increased in H group, whereas *Carnobacterium* and *Brucella* increased
in C group (Figure 4A).

At the same time, we also compared the differences between H and C groups of
bacteria obtained by culturomics. We drew a heat map of the proportion of each
bacterium in the total sample. *Streptococcus orails*, *Veillonella ayptica*, and
*Parvimonas micra* were both cultured at a high frequency without significant group
differences. *Prevotella orails* was found with a significantly higher frequency in H
group ($P=0.019$) (Figure 4B). This was coincident with sequencing result, which also
showed the prevalent of *Prevotella* in H group.

**5, Different pathological types shaped different pulmonary** 243 **microbiomes**

We then sought to disclose differences in lung microbiota of NSCLC and SCLC.
Basic information included age, sex, smoking status and distant metastasis were
comparable between 2 groups (Table S2). Bacteria with abundance greater than 4%
were considered as dominant. Across the 32 NSCLC lung samples, the dominant
genera were *Pseudomonas* (32%), *Streptococcus* (5%), and *Bacillus* (4%). While in
SCLC, the dominant genus was *Pseudomonas* (31%), *Veillonella* (7%), and
*Prevotella_7* (6%) (Figure 5A). NSCLC was not significantly different from SCLC in
α -diversity, as measured by Shannon ($P=0.931$) and Chao1 diversity index ($P=0.720$),
or β -diversity as measured by Bray-Curtis distances ($P=0.489$) (Figure S4 A, B). By
culturomics study, we found that 77 species were only isolated from NSCLC lung
samples and 37 species were only isolated from SCLC lung samples (Figure 5B).
Only the bacterial species isolated at least half of the patients in the group were
selected to the next step of comparison. Then we analyzed bacteria, which only
isolated from one group with isolation rate $\geq 50\%$ (The bacterial species isolated from
at least half of the patients in the group). *Gemella sanguinis*, *Pseudoramibacter*
*alactolyticus*, *Bifidobacterium dentium*, and *Streptococcus intermedius* were the four
species with a relatively high frequency only isolated from the NSCLC group. Of all
the species isolated from SCLC lung samples, *Prevotella pallens* was the species
which had a relatively high frequency. Then, we compared the relative abundance of

above five bacteria at genus level with 16S rDNA amplicon sequencing. Though there
is no significant difference between the two groups, the abundance showed a higher
tendency in the only isolated site (Figure 5C).

In subtypes analysis of NSCLC patients, among the 16 ADC samples, the dominant
genera were *Pseudomonas* (26%) and *Bacillus* (8%). While in 14 SCC samples, the
dominant genera were *Pseudomonas* (38%) and *Streptococcus* (8%) (Figure 5D).
Shannon ($P=0.270$) and Chao1 diversity index ($P=0.402$) were not significantly
different in ADC and SCC groups, while the β -diversity showed that microbiota
constitution in ADC lung samples was clearly different than the SCC samples
($P=0.021$) (Figure S4 C, D). In the LEfSE analysis of ADC and SCC, 11 various taxa
were detected to display contrasting correlations between NSCLC subtypes. A
differential abundance analysis at the genus level between ADC and SCC showed an
enrichment of *Bacillus* and *Castellaniella* in ADC patients, whereas SCC had a higher
abundance of *Brucella* (Figure 5E).

Discussion

To date, criteria for the normal composition of lung microbiota have not yet been
established, but the available data indicate that their composition in cancer patients
differs considerably from that of healthy individuals [26]. An association between the
lung microbiome and histologic classification of lung cancer was also observed.
*Pseudomonas* showed a correlation with ADC [27] interestingly, the bacteria found
in BALF can originate in the mouth [28]. With more oral microbes entering to the
lungs being associated with increased lung proinflammatory cytokines [29]. The
microbiome has been implicated in lung cancer in a variety of specific ways, but the
role of lung microbiota in carcinogenic processes has not yet been elucidated. One of
the possible reasons for this is the current studies are mostly based on the sequencing
levels and resulting in a lack of strains materials. In the current study, both
culturomics and 16S rDNA amplicon sequencing were employed to evaluate and
compare the structure and diversity characteristics of oral and pulmonary microbiota
associated with lung cancer.

**Culturomics** can identify bacteria to strain level, and successfully isolating living
bacteria is crucial for carrying out subsequent experimental work [15, 30, 31]. A study
summarized a list of microbes isolated from the human lower respiratory tract showed

that the lung microbiome was dominated by the phyla *Pseudomonadota*, *Firmicutes*,
*Bacteroidota*, and *Actinomycetota*, which was consistent with our study [19]. In this
study, we have cultured 198 identified bacterial species from the human BALF and
156 from oral of lung cancer patients, and 15 potentially new taxa. The related article
about novel bacteria new.4 and new.10 which belonged to a novel genus we named
‘*Curtanaerobium*’ was submitted to IJSEM (under review). The present study enabled
to expand of the human respiratory and oral repertoire. We also identified 20 species
as prevalent strains in both BLAF and oral samples of lung cancer patients, which
indicated the pulmonic microbiota maybe come from oral cavity. Cultruomics could
reduce the number of these unclassified or no-rank OTUs by increasing the number of
pure cultured microorganism species. In our study, *Pseudoramibacter alactolyticus*
was recovered with a relatively high frequency only isolated from the NSCLC group
but was not detected in 16S rDNA amplicon sequencing. *Parvimonas micra* was
reported to reveal a high abundance in colorectal cancer patients, and Yu et al.
reported that *P. micra* promoted colorectal tumorigenesis by inducing colonocyte
proliferation and altering Th17 immune response [32, 33]. Still, there was no report
of the correlation with lung cancer. In this study, *P. micra* was cultured at a high
frequency in lung cancer patients, and the roles of these strains provided materials for
further study.

Microbiota composition analysis at the genus level through taxonomic assignment
were performed, and the results showed that representative flora differed by sampling
site. We found that *Streptococcus*, *Veillonella*, and *Prevotella* were enriched in the
oral samples, while *Pseudomonas* was enriched in BLAF samples. *Streptococcus* and
*Veillonella* which were considered as oral commensals were reported to increase in
lower airways of lung cancer patients [34]. *Pseudomonas* species are commonly
found in the respiratory tract; and are involved in the pathogenesis of lung diseases,
such as chronic obstructive pulmonary disease and cystic fibrosis [35, 36]. One study
reported that *Pseudomonas aeruginosa* was significantly more abundant in brain
metastasis of NSCLC patients [37]. Likewise, the majority of patients in our study
was with distant metastasis, and *Pseudomonas* was the most genus in BALF samples.
These findings, including ours, seem to suggest that *Pseudomonas* colonize more
frequently in the respiratory tract of lung cancer patients and induce a promotion of
distant metastasis.

*Gemella sanguinis*, *Pseudoramibacter alactolyticus*, and *Streptococcus intermedius*
are involved in human inflammatory diseases, such as endocarditis [38], periodontal
infectious [39], and brain abscess [40]. Germ-free mice or mice treated with
antibiotics had a significantly lower incidence of lung cancer than pathogen-free
mice [41, 42]. Our results showed that the three species were only isolated from lung
microbiota of NSCLC, not isolated from SCLC patients, and the tendency was
confirmed by 16S rDNA amplicon sequencing, which indicated that the specific
bacterial might mediate the development of NSCLC by inducing chronic
inflammation.

When stratifying by pathological subtypes of lung cancer, a significant enrichment in
*Bacillus* and *Castellaniella* was observed in the BALF samples of ADC patients, and
an enrichment of *Brucella* was observed in the SCC group. Another study reported the
enrichment of *Acinetobacter* in BALF samples of ADC, while *Bacillus* and *Brucella*
showed no significant differences between ADC and SCC groups [11]. These
conflicting associations of microbiome with different pathological subtypes could be
attributable to the significant variation across different individuals and sampling
methods.
In conclusion, we examined pneumonic and oral microbiota in lung cancer patients
using cultuomics and 16S rDNA sequencing, and found that the microbial
community changed in lung cancer patients, which diversity might be site- and
pathological-dependent.  We found that *Streptococcus* and *Veillonella* were the highly
dominant bacteria both in both pneumonic and oral samples of lung cancer patients by
**cultuomics**, which suggested possible deleterious effects of airway microbial
dysbiosis originating from oral cavity. We showed that *Prevotella orails* was only
isolated from H group and *Gemella sanguinis* was only isolated from NSCLC group,
which was consistent with the 16S rDNA amplicon sequencing. This study provides
basic data on the microbiota diversity in pneumonic and oral samples from lung
cancer patients. These features may be potential bacterial biomarkers and new targets
for lung cancer diagnosis and treatment and the isolated strains provide materials for
exploring the causative relationships.

**AUTHOR CONTRIBUTIONS**

**Ethics approval and consent to participate**

All Procedures performed in studies involving human participants or animals were
approved by the Institutional Review Board of the Peking University School of
Oncology (No. 2018KT89). All patients gave their written consent before inclusion in
the study.

**Consent for publication**

Not applicable.

**Availability of data and materials**

All the data generated or analyzed in this study are included in this published article
(or its Supplementary Information files). The read sequences obtained from Illumina
NovaSeq were submitted to the NCBI Sequence Read Archive (SRA) under accession
number PRJNA904049 (BioProject ID)
(<http://www.ncbi.nlm.nih.gov/bioproject/?term=PRJNA904049>).

**Competing Interest**

The authors declare no competing interests.

**Funding**

This research was supported by the National Key Research and Development
Program of China (2021YFC2301000), and the National Natural Science Foundation
of China (31970863 and 81790632).

**Authors' contributions**

Y.J.B. designed research and project outline. R.F.Y and Z.P.W. directed the research.
Y.F.S, J.J.L., Y.J.L., A.T.T, Z.M.L, M.M.L, J.B, and Z.H.W. performed isolation,
deposition and identification. Y.F.T. and Z.Y.P. performed genome analysis. Y.F.S.
and Y.J.B. drafted the manuscript. All authors read and approved the final manuscript.

**Acknowledgements**

We gratefully acknowledge all of the participants and the medical staff who helped
collect the samples in this study.

Reference

- 1. Murray CJ, Lopez AD. Alternative projections of mortality and disability by cause 1990-
2020: Global Burden of Disease Study. *Lancet*. 1997;349(9064):1498-504.
- 2. Saus E, Iraola-Guzmán S, Willis JR, Brunet-Vega A, Gabaldón T. Microbiome and
colorectal cancer: Roles in carcinogenesis and clinical potential. *Mol Aspects Med*. 2019;69.
- 3. Hanahan D. Hallmarks of Cancer: New Dimensions. *Cancer Discov*. 2022;12(1):31-46.
- 4. Dong Q, Chen ES, Zhao C, Jin C. Host-Microbiome Interaction in Lung Cancer. *Frontiers*
*In Immunology*. 2021;12:679829.
- 5. Zheng L, Sun R, Zhu Y, Li Z, She X, Jian X, et al. Lung microbiome alterations in NSCLC
patients. *Scientific Reports*. 2021;11(1):11736.
- 6. Lee SH, Sung JY, Yong D, Chun J, Kim SY, Song JH, et al. Characterization of
microbiome in bronchoalveolar lavage fluid of patients with lung cancer comparing with
benign mass like lesions. *Lung Cancer*. 2016;102:89-95.
- 7. Liu Y, O'Brien JL, Ajami NJ, Scheurer ME, Amirian ES, Armstrong G, et al. Lung tissue
microbial profile in lung cancer is distinct from emphysema. *Am J Cancer Res*.
2018;8(9):1775-87.
- 8. Zhuo M, An T, Zhang C, Wang Z. Characterization of Microbiota in Cancerous Lung and
the Contralateral Non-Cancerous Lung Within Lung Cancer Patients. *Front Oncol*.
2020;10:1584.
- 9. Liu H-X, Tao L-L, Zhang J, Zhu Y-G, Zheng Y, Liu D, et al. Difference of lower airway
microbiome in bilateral protected specimen brush between lung cancer patients with unilateral
lobar masses and control subjects. *Int J Cancer*. 2018;142(4):769-78.

- 10. Huang D, Su X, Yuan M, Zhang S, He J, Deng Q, et al. The characterization of lung
microbiome in lung cancer patients with different clinicopathology. *Am J Cancer Res.*
2019;9(9):2047-63.
- 11. Gomes S, Cavadas B, Ferreira JC, Marques PI, Monteiro C, Sucena M, et al. Profiling of
lung microbiota discloses differences in adenocarcinoma and squamous cell carcinoma.
*Scientific Reports.* 2019;9(1):12838.
- 12. Christiani DC. The oral microbiome and lung cancer risk. *Thorax.* 2021;76(3):216-7.
- 13. Tsay J-CJ, Wu BG, Sulaiman I, Gershner K, Schluger R, Li Y, et al. Lower Airway
Dysbiosis Affects Lung Cancer Progression. *Cancer Discov.* 2021;11(2):293-307.
- 14. Yang J, Mu X, Wang Y, Zhu D, Zhang J, Liang C, et al. Dysbiosis of the Salivary
Microbiome Is Associated With Non-smoking Female Lung Cancer and Correlated With
Immunocytochemistry Markers. *Front Oncol.* 2018;8:520.
- 15. Bilen M, Dufour J-C, Lagier J-C, Cadoret F, Daoud Z, Dubourg G, et al. The contribution
of culturomics to the repertoire of isolated human bacterial and archaeal species. *Microbiome.*
2018;6(1):94.
- 16. Lagier J-C, Dubourg G, Million M, Cadoret F, Bilen M, Fenollar F, et al. Culturing the
human microbiota and culturomics. *Nature Reviews Microbiology.* 2018;16:540-50.
- 17. Chang Y, Hou F, Pan Z, Huang Z, Han N, Bin L, et al. Optimization of Culturomics
Strategy in Human Fecal Samples. *Frontiers In Microbiology.* 2019;10:2891.
- 18. Chun J, Oren A, Ventosa A, Christensen H, Arahal DR, da Costa MS, et al. Proposed
minimal standards for the use of genome data for the taxonomy of prokaryotes. *Int J Syst*
*Evol Microbiol.* 2018;68(1):461-6.

- 19. Fonkou MD, Dufour J-C, Dubourg G, Raoult D. Repertoire of bacterial species cultured
from the human oral cavity and respiratory tract. *Future Microbiol.* 2018;13:1611-24.
- 20. Lagier J-C, Khelaifia S, Alou MT, Ndongo S, Dione N, Hugon P, et al. Culture of
previously uncultured members of the human gut microbiota by culturomics. *Nature*
*Microbiology.* 2016;1:16203.
- 21. Dubourg G, Morand A, Mekhalif F, Godefroy R, Corthier A, Yacouba A, et al.
Deciphering the Urinary Microbiota Repertoire by Culturomics Reveals Mostly Anaerobic
Bacteria From the Gut. *Frontiers In Microbiology.* 2020;11:513305.
- 22. Morand A, Cornu F, Dufour J-C, Tsimaratos M, Lagier J-C, Raoult D. Human Bacterial
Repertoire of the Urinary Tract: a Potential Paradigm Shift. *Journal of Clinical Microbiology.*
2019;57(3).
- 23. Price TK, Dune T, Hilt EE, Thomas-White KJ, Kliethermes S, Brincat C, et al. The
Clinical Urine Culture: Enhanced Techniques Improve Detection of Clinically Relevant
Microorganisms. *Journal of Clinical Microbiology.* 2016;54(5):1216-22.
- 24. Kitagawa K, Shigemura K, Onuma K-I, Nishida M, Fujiwara M, Kobayashi S, et al.
Improved bacterial identification directly from urine samples with matrix-assisted laser
desorption/ionization time-of-flight mass spectrometry. *J Clin Lab Anal.* 2018;32(3).
- 25. Letunic I, Bork P. Interactive Tree Of Life (iTOL) v5: an online tool for phylogenetic tree
display and annotation. *Nucleic Acids Res.* 2021;49(W1):W293-W6.
- 26. Mao Q, Jiang F, Yin R, Wang J, Xia W, Dong G, et al. Interplay between the lung
microbiome and lung cancer. *Cancer Letters.* 2018;415:40-8.
- 27. Kovaleva OV, Romashin D, Zborovskaya IB, Davydov MM, Shogenov MS, Gratchev A.

Human Lung Microbiome on the Way to Cancer. *Journal of Immunology Research*.
2019;2019:1394191.

28. Venkataraman A, Bassis CM, Beck JM, Young VB, Curtis JL, Huffnagle GB, et al.
Application of a neutral community model to assess structuring of the human lung microbiome.
*mBio*. 2015;6(1).

29. Zhang J, Wu Y, Liu J, Yang Y, Li H, Wu X, et al. Differential Oral Microbial Input
Determines Two Microbiota Pneumo-Types Associated with Health Status. *Adv Sci (Weinh)*.
2022;9(32):e2203115.

30. Bilen M. Strategies and advancements in human microbiome description and the
importance of culturomics. *Microb Pathog*. 2020;149:104460.

31. Bellali S, Lagier J-C, Million M, Anani H, Haddad G, Francis R, et al. Running after
ghosts: are dead bacteria the dark matter of the human gut microbiota? *Gut Microbes*.
2021;13(1).

32. Zhao L, Zhang X, Zhou Y, Fu K, Lau HC-H, Chun TW-Y, et al. *Parvimonas micra*
promotes colorectal tumorigenesis and is associated with prognosis of colorectal cancer
patients. *Oncogene*. 2022;41(36):4200-10.

33. Löwenmark T, Löfgren-Burström A, Zingmark C, Eklöf V, Dahlberg M, Wai SN, et al.
*Parvimonas micra* as a putative non-invasive faecal biomarker for colorectal cancer. *Scientific*
*Reports*. 2020;10(1):15250.

34. Tsay J-CJ, Wu BG, Badri MH, Clemente JC, Shen N, Meyn P, et al. Airway Microbiota Is
Associated with Upregulation of the PI3K Pathway in Lung Cancer. *Am J Respir Crit Care*
*Med*. 2018;198(9):1188-98.

- 35. Jurado-Martín I, Sainz-Mejías M, McClean S. : An Audacious Pathogen with an
Adaptable Arsenal of Virulence Factors. *International Journal of Molecular Sciences*.
2021;22(6).
- 36. Biswas L, Götz F. Molecular Mechanisms of and Interactions in Cystic Fibrosis. *Frontiers*
*In Cellular and Infection Microbiology*. 2021;11:824042.
- 37. Lu H, Gao NL, Tong F, Wang J, Li H, Zhang R, et al. Alterations of the Human Lung and
Gut Microbiomes in Non-Small Cell Lung Carcinomas and Distant Metastasis. *Microbiol*
*Spectr*. 2021;9(3):e0080221.
- 38. Maraki S, Plevritaki A, Kofteridis D, Scoulica E, Eskitzis A, Gikas A, et al. Bicuspid aortic
valve endocarditis caused by *Gemella sanguinis*: Case report and literature review. *J Infect*
*Public Health*. 2019;12(3):304-8.
- 39. Antunes HS, Rôças IN, Alves FRF, Siqueira JF. Total and Specific Bacterial Levels in the
Apical Root Canal System of Teeth with Post-treatment Apical Periodontitis. *J Endod*.
2015;41(7):1037-42.
- 40. Issa E, Salloum T, Tokajian S. From Normal Flora to Brain Abscesses: A Review of.
*Frontiers In Microbiology*. 2020;11:826.
- 41. Yadava K, Pattaroni C, Sichelstiel AK, Trompette A, Gollwitzer ES, Salami O, et al.
Microbiota Promotes Chronic Pulmonary Inflammation by Enhancing IL-17A and
Autoantibodies. *Am J Respir Crit Care Med*. 2016;193(9):975-87.
- 42. Jin C, Lagoudas GK, Zhao C, Bullman S, Bhutkar A, Hu B, et al. Commensal Microbiota
Promote Lung Cancer Development via $\gamma\delta$ T Cells. *Cell*. 2019;176(5).

**Fig 1.** Summary of culturomics methods and workflow.

**Fig 2.** The bacteria identified from the BALF samples. **A** Phylogenetic tree of the
isolated bacterial species from BALF samples. **B** Proportion of bacterial species
isolated from the BALF samples listed according to their phylum. **C** Upsetplot
showing the shared cultured species between human oral/respiratory, gut, urine and
vagine.

**Fig 3.** The microbial composition varied in different anatomy sites. **A** Taxonomic
composition at phylum level in BALF and oral samples based on 16S rDNA amplicon
sequencing (Top 30). **B** The Shannon diversity index of BALF samples and oral. *P*
values were calculated with Wilcoxon test. **P* < 0.05, ***P* < 0.01 **C** PCoA analysis of
three different anatomy sites samples in lung cancer patients. Anosim was performed
to test statistically whether there is a significant difference. ***P* < 0.01 **D** Box-and-
whisker plots illustrating the differences of the significantly 2 genus relative
abundances in three different anatomy sites. * *P* < 0.05 **E** Proportion of bacterial
species isolated from C, H and O samples listed according to their phylum. **F** Veen
diagram of culturable bacterial species in three different anatomy sites. C: samples
from the cancerous site of lung cancer patients, H: samples from the contralateral
“healthy” controls of lung cancer patients. O: samples from oral site of lung cancer
patients.

**Fig 4.** Differentially abundant taxonomy between cancer and paired “healthy” lung. **A**
The LEfSE was used to identify the bacterial microbiota that significantly differed

between cancer and lung. Only taxa meeting a significant LDA threshold value of >

3.5 and *P* < 0.05 are shown. **B** Heatmap analysis between C and H of lung microbiota
was done based on the culturomics of the cancer patients (n=15). Each row represents
an individual species with the higher isolated rate in the two groups. The top 25
bacterial species recovered from either group. From blue to red represents the
frequency of culturomics recovery from 0 to 1. Statistical comparisons were made
using the Fischer Exact test with FDR correction. **P* < 0.05

**Fig 5.** Characterization of the lung microbiota in the NSCLC and SCLC and the
differences in ADC and SCC. **A** Taxonomic composition at genera level in NSCLC
and SCLC lung samples. **B** Veen diagram of culturable bacterial species only isolated
from NSCLC and only isolated from SCLC lung samples. **C** Difference of relative the
genus abundance corresponding to species isolated from NSCLC and SCLC lung. To
identify differential microbial taxa, paired t-test was performed and *P*-values were
adjusted for multiple comparison by the FDR. **D** Taxonomic composition at genera
level in ADC and SCC lung samples. **E** The LEfSE was used to identify the bacterial
microbiota that significantly differed between NSCLC and SCLC. Only taxa meeting
a significant LDA threshold value of > 4 and *P* < 0.05 are shown.

**Table 1.** Clinical characteristics of patients.

Variable	NSCLC	SCLC	Non
N	16	7	2
Age-mean (SD)	65 (7.0)	67 (12.8)	69 (8.2)
Gender			
Male, n (%)	9 (56%)	5 (72%)	2
Female, n (%)	7 (44%)	2 (14%)	0
Smoking			
Current or former Smoker, n (%)	12(75%)	6(86%)	1
Never smoker, n (%)	6(25%)	1(14%)	1
Pathological diagnosis			
Adenocarcinoma, n (%)	8(50%)	—	—
Squamous cell carcinoma, n (%)	7(44%)	—	—
Unidentified	1(6%)	—	—
Distant			
MO	4(25%)	2(29%)	—
M1	12(75%)	5(71%)	—

**Table 2.** Summary of the proportion of the TOP20 bacterium in BALF samples and
 the related proportion in oral cavity.

Genera	Species	C	H	O
	Streptococcus salivarius	60.00%	60.00%	73.33%
	Streptococcus pseudopneumoniae	53.33%	40.00%	33.33%
	Streptococcus parasanguinis	93.33%	86.67%	100.00%
Streptococcus	Streptococcus oralis	93.33%	93.33%	100.00%
	Streptococcus mitis	93.33%	86.67%	100.00%
	Streptococcus gordonii	53.33%	53.33%	80.00%
	Streptococcus constellatus	66.67%	73.33%	93.33%
	Streptococcus anginosus	66.67%	66.67%	100.00%
Veillonella	Veillonella parvula	60.00%	60.00%	73.33%
	Veillonella atypica	66.67%	80.00%	66.67%
Solobacterium	Solobacterium moorei	66.67%	60.00%	73.33%
Slackia	Slackia exigua	60.00%	73.33%	60.00%
Parvimonas	Parvimonas micra	80.00%	80.00%	93.33%
Mogibacterium	Mogibacterium diversum	53.33%	46.67%	53.33%
Granulicatella	Granulicatella adiacens	66.67%	53.33%	46.67%
Gemella	Gemella morbillorum	66.67%	53.33%	66.67%
Fusobacterium	Fusobacterium nucleatum	40.00%	66.67%	80.00%
Dialister	Dialister invisus	73.33%	53.33%	73.33%
Anaeroglobus	Anaeroglobus geminatus	53.33%	60.00%	73.33%
Actinomyces	Actinomyces odontolyticus	80.00%	80.00%	66.67%

553 **C:** samples from the cancerous site of lung cancer patients, **H:** samples from the contralateral
 554 “healthy” controls of lung cancer patients. **O:** samples from oral site of lung cancer patients.
 555

A

B

C

A

B

C

D

E

F

A

B

A

B

C

D

E

Response to Reviewer comments

Reviewer #10 (Comments for the Author):

The manuscript titled, "Characterization of lung and oral microbiome in lung cancer patients by culturomics and 16S rDNA sequencing" by Sun et al provides valuable insight about the bacterial populations in the lung and oral bacteriome in lung cancer patients. Overall, I firmly believe the study provides important information which can be translated to future animal models, and also pave the way for studies that may use the identified bacterial species as biomarkers. The methods are also well explained and the bioinformatic tools were utilized well in the study. However, I have some major and minor comments for the authors to consider. I believe the manuscript will be much improved if the authors incorporate my suggestions.

Major comments

1. The authors use the word "culturomics" throughout the Text, but I believe the word is "culturomics". Is it a typo throughout the manuscript? I think the authors must address this point, because the word is a part of the manuscript title.

Response: We apologize for the mistake. We agree with the reviewer's comment, and we have thoroughly checked and revised the manuscript.

2. I firmly believe the manuscript requires extensive English editing. I have come across numerous grammatical errors and typos. I have tried to highlight most of them in the minor comments, but still, I have the feeling that the text requires further editing.

Response: We apologize for the poor language of our manuscript. We have now worked on the language and have also involved native English speakers for language corrections.

3. In the discussion section, I think the authors should add some information about the bacterial species that were identified in the cancer patients. For instance, why *Pseudomonas* is such a potential threat in lung cancer, is it due to certain pathogenicity factors that are encoded by its genome? Similarly, I think adding some information about the other identified species will provide more rigor to the Discussion section.

Response: Thanks for the suggestion. We have added the information about the differential bacteria at species level in the discussion section, including *Pseudomonas aeruginosa* and *Parvimonas micra*, and *Brucella melitensis*. And we think combined the discussion of these species and our findings will provide a deeper insight for understanding the links between the species and the lung cancer.

4. Minor comments

- Line 25, mention the full name of BALF not just the abbreviation
- Line 26, "Twenty" not 20.
- Line 27, "BALF" instead of "BLAF".
- Line 31, mention the full name of "LEfSE".
- Line 33, "Prevotella Oralis" not "Prevotella orails"
- Line 39, "found" not "founded".
- Line 41, add "this study" before "provides"

- Line 56-58, should be rephrased.
- Line 67, add "an" instead of "the"
- Line 68 has some spacing issues.
- Line 81 again "BLAF".
- Line 87, add a full stop after 1
- Line 93, "Twenty five" instead of "25"
- Line 93, I think "from patients" is a mistake.
- Line 95, the word should be "transbronchoendoscopy"
- Line 98-99, replace ", total 21 patients' oral samples were collected" with "oral samples from 21 patients were collected".
- Line 113, "100 μ L" not "100ul"
- Line 116, mention the full name of "MALDI-TOF MS".
- Line 118 has a strange spacing kindly check.
- Line 141 "raw" instead of "Raw".
- Line 143 add "were" after "sequences"
- Line 171, replace "were not" with "did not".
- Line 176, check if the word "cultruoims" is correct?
- Line 182, I think the exact values should be mentioned that are given in the Figure 2A.
- Line 188, you mention that 156 bacterial species were identified, but there is not mention of this in the Supplementary information.
- Line 204-205, its hard to understand the meaning, kindly rephrase.
- Line 220, add "of" after "prevalence"
- Line 240, "results" instead of "result"
- Line 241, "prevalence" not "prevalent"
- Line 242 , "Different pathological types shaped different pulmonary microbiomes" the name should be changed to "Different pathological strains reshape different pulmonary microbiomes"
- Line 318, again "BLAF"
- Line 325, add "prevalent" before "genus".
- Line 331, there is a double space before "Germ-free".
- Line 336, "bacteria" not "bacterial".
- Line 363, "procedures" not "Procedures".

Response: We are grateful to the reviewer for his remarks, and we have corrected all these errors in the revised manuscript.

Reviewer #12 (Comments for the Author):

The objective of this research aimed to analyze the lung and oral microbiome of patients with lung cancer.

1. There is a lack of information concerning the patients (oral pathologies?).

Response: Thanks for the suggestion. As our study focused on the microbiomes (both oral and pulmonary) of the lung cancer patients, we took into account the oral condition when setting the criteria for patient recruitment. The inclusion criteria had been added in the revised manuscript.

2. The conclusions must be accompanied by several reservations concerning the size of the sample, the natural biodiversity of the microbiome between patients.

Response: We thank the reviewer for his suggestion. We agree that the presence of natural biodiversity of the microbiome across individuals. We have added the reservations concerning the size of the sample in the discussion section as suggested . And we will call for larger and dynamic longitudinal studies in the future to verify the association between microbiome and lung cancer.

3. Comparing the microbiomes (oral and pulmonary) of the same patient from a cancerous lung area with another non-cancerous area of the same patient cannot be 100% reliable.

Response: We thank the reviewer for his relevant comment. We admit that spatial variation of microbiota within an individual is significantly less than variation across individuals in healthy lungs. However, the local tumor microenvironment of cancerous lung may be changed. We assume that the microbial differences between the cancerous lung and “healthy lung” may be associated with the tumor initiation and development. To date, two studies used high-throughput sequencing for characterizing the difference in microbiota compositions between cancerous and “healthy” lung. Zhuo et al found that the relative abundance of family Spiroplasmataceae, and its genus *Spiroplasma* was significantly increased in cancerous lung (Zhuo, M., et al., *Characterization of*

Microbiota in Cancerous Lung and the Contralateral Non-Cancerous Lung Within Lung

Cancer Patients. *Frontiers In Oncology*, 2020. 10: p. 1584). Liu et al found that the lower

airway microbial communities of lung cancer were distinguishable from that of “healthy”

controls (Liu, H.-X., et al., *Difference of lower airway microbiome in bilateral protected*

specimen brush between lung cancer patients with unilateral lobar masses and control

subjects. *International Journal of Cancer*, 2018. 142(4): p. 769-778).

4 In the study, a significant segregation was found at the level of the genus, but without having sought the equivalent level at the species. The taxonomic levels of the genus are not the relevant biological measurement units for some authors (BIK, ME et al. Bacterial diversity in the oral cavity of 10 healthy individuals) International society for microbial ecology. (2010). However, in the mouth, the level of ecological interest chosen to label is the genus rather than species. There also reservations concerning the conclusions of the article are to be made.

Response: We are grateful to the reviewer for his suggestion. As the reviewer appointed, the 16S rDNA sequencing gave the bacterial identification at genus level, which is relatively accurate, as we compared to the results of culturomics, which gave the identification at species level. We simply compared them at genus level, indicating that only equivalent genus abundance corresponding to species obtained by the culturomics was considered. We have discussed this issue in the discussion section, according to the reviewer's suggestion, pointing out the limitation of this research.

4. Add Bibliography.

Ramirez-Labrada, AG et al. The influence of microbiota on lung carcinogenesis immunity and immunotherapy. Trends Cancer 2020. 6, 86-97.

Yagi, K.; Huffnagle, G.B.; Lukacs, N.W.; Asai, N. The Lung Microbiome during Health and Disease. Int. J. Mol. Sci. 2021, 22, 10872. <https://doi.org/10.3390/ijms221910872>

Dickson, R.P.; Erb-Downward, J.R.; Martinez, F.J.; Huffnagle, G.B. The Microbiome and the Respiratory Tract. Ann. Rev. Physiol. 2016, 78, 481-504. [CrossRef]

Pizzo F. et al. Role of the microbiota in lung cancer insights on prevention and treatment. Int Mol Sci. 2022 23. (11) 6138.

Q. et al. Alterations of fecal bacterial communities in patients with lung cancer. Am J Transl 2018. 10 3171 -3185.

Georgiou K et al . Gut microbiota and lung cancer. Where Do we stand? Int J mol Sci 2021.

Goubet AG. Et al The impact of the intestinal microbiota in therapeutic responses against cancer. Comptes Rendus Biol 2018. 341 284-289.

Segal LN et al. enrichment of the lung microbiome with oral taxa is associated with lung inflammation of Th17 phenotype. Nat Microbiol 2016.

Mortensen MS. The developing hypopharyngeal microbiota in early life. Microbiome 2016.

Das S. et al . A prevalent and culturable microbiota links ecological balance to clinical stability of the human lung after transplantation. Nat Commun 2021.

Response: Thanks for the suggestion. We have carefully read the bibliographies and added them in the manuscript.

Reviewer #20 (Comments for the Author):

This is an interesting study aimed at characterizing and associating the oral and pulmonary microbiome of lung cancer patients. However, the study has not considered that the oral microbiome depends mainly on the oral condition. It has been widely that high bacterial richness in the salivary microbiota is significantly associated with poor oral health, as indicated by decayed teeth, periodontitis, and poor oral hygiene. Therefore, it becomes crucial to understand oral microbial diversity and how it fluctuates under conditions of disease/disturbance. Advances in metagenomics and next-generation sequencing techniques generate rapid sequences and provide extensive information on the microorganisms inhabiting a niche. Therefore, the information retrieved can be used to develop microbiome-based biomarkers for use in the early diagnosis of oral and associated diseases. However, a homogenization of the oral clinical conditions of the samples must be considered for the results to be robust.

Response: Thanks for the suggestion. As our study focusing on the microbiomes (oral and pulmonary) of the lung cancer patients, we took into account the oral condition when setting the criteria for patient recruitment. All enrolled patients had not recently suffered from oral disease. We have added the inclusion criteria in the patient recruitment and samples collection section.

Reviewer #5 (Comments for the Author):

1. p should be represented properly in the manuscript as it is a statistical symbol.

Response: We are grateful to the reviewer for his remark. We have corrected in the manuscript.

2. Notch box plot should be used instead of box whisker plot.

Response: We appreciate the reviewer's suggestion. We have redesigned the figures as suggested.

3. The no of clinical samples may be increased.

Response: Thanks for the suggestion. We have added the reservations concerning the size of the sample in the discussion section as suggested. And we will call for larger and dynamic longitudinal studies in the future to verify the association between microbiome and lung cancer.

4. The authors should also comment on the keystone species and species diversity of each microbiome and their changes.

Response: Thanks for the suggestion. We have added the information and changes of the keystone species in the discussion section, including *Pseudomonas aeruginosa*, *Parvimonas micra*, and *Brucella melitensis*.

April 3, 2023

Dr. Yujing Bi
Academy of Military Medical Sciences State Key Laboratory of Pathogen and Biosecurity
Beijing
China

Re: Spectrum00314-23R1 (Characterization of lung and oral microbiomes in lung cancer patients using culturomics and 16S rDNA sequencing)

Dear Dr. Yujing Bi:

Your manuscript has been accepted, and I am forwarding it to the ASM Journals Department for publication. You will be notified when your proofs are ready to be viewed.

Sincerely,

Diyang Li
Editor, Microbiology Spectrum

Journals Department

Characterization of lung and oral microbiomes in lung cancer

patients using culturomics and 16S rDNA sequencing

Yifan Sun^{1,#}, Yuejiao Liu^{1,#}, Jianjie Li², Yafang Tan¹, Tongtong An², Minglei Zhuo²,

Zhiyuan Pan¹, Menglei Ma², Bo Jia², Hongwei Zhang², Ziping Wang^{2,*}, Ruifu Yang^{1,*},

Yujing Bi^{1,*}

[revised manuscript text omitted]

rDNA gene was amplified using specific primers (341F: 5'-
CCTAYGGGRBGCASCAG-3'; 806R: 5'-GGACTACNNGGGTATCTAAT-3').
Sequencing libraries were generated using TruSeq® DNA PCR-Free Sample
Preparation Kit (Illumina, USA) following manufacturer's recommendations and
index codes were added. The library quality was assessed on the Qubit® 2.0
Fluorometer (Thermo Scientific) and Agilent Bioanalyzer 2100 system. The
sequencing was sequenced on an Illumina Novaseq6000 sequencing platform
(Illumina, San Diego, CA, United States) and 250 bp paired-end reads were
generated.

**4. Sequence data analysis**

Raw reads were filtered to remove adaptors, low-quality reads, and ambiguous bases.
Clean data was extracted from the raw data using USEARCH 8.0. The operational
taxonomic units (OTUs) were classified based on 97% similarity after chimeric
sequences were removed using UPARSE (version 7.0.1001 <http://drive5.com/uparse/>),
and the representative sequence from each OTU cluster was obtained. α diversity was
assessed using the nonparametric Shannon and Simpson indices. β diversity was
calculated using Bray–Curtis distances in QIIME and visualized by principal
coordinate analysis (PCoA). Linear discriminant analysis effect size (LEfSe) was used
to detect taxa with differential abundance among groups. The Metastats analysis was
employed to detect the differences in the microbiota composition at the genus level.

**5. Statistical analyses**

Statistical analyses were performed using the R software (v3.4.10) and SPSS 20.0.
Independent t- and chi-square tests were used to analyze the demographic and clinical
data. Fischer's exact test with false discovery rate (FDR) correction was used to
compare frequency of isolated species between groups. Sample diversity metrics were
assessed based on the nonparametric Shannon index and Chao1 index. PCoA plots
were generated to visualize the separation of samples based on pairwise distances, and
ANOSIM was performed to evaluate the statistically significant difference in PCoA
analysis. p -value < 0.05 was considered statistically significant.

**RESULTS**

**1. Clinical information**

We recruited 25 patients with unilateral lobar masses from Beijing Peking University
Cancer Hospital between January 2021 and May 2022, of which 23 were newly
diagnosed with lung cancer via histological confirmation but 2 were not. There were
16 non-small-cell lung cancer (NSCLC) patients, including 8 ADC, 7 SCC, and one
with non-specified NSCLC, and 7 small-cell lung cancer (SCLC) patients. None
previously had received any anticancer therapy, radiation therapy or any antibiotics
treatment. Of the 23 patients with lung cancer, 14 were men and 9 were women.

Furthermore, 18 were smokers. There were 17 and 6 patients with and without distant
metastasis, respectively (Table 1).

**2. Characteristics of bacteria isolated from the lung and oral cavity** 188 **via culturomics**

Figure 1 shows the culturomics workflow. Briefly, total 45 samples were collected
from 15 lung cancer patients, and four culture conditions were tested for each sample.
We obtained 12379 colonies from the BALF samples (both C and H samples), and
198 bacteria were identified at the species level using MALDI-TOF or 16S rDNA
gene sequencing. The identified species belonged to six phyla, including Firmicutes
(39.90%), Proteobacteria (27.78%), Actinobacteria (19.19%), Bacteroidetes (9.60%),
Fusobacteria (2.02%), and Synergistetes (1.52%) (Figures 2A and B). Comparison
with the previously established repertoire of microorganisms isolated from the human
gut, urine, vagina, and oral/respiratory tract revealed that approximately 1/4th of the
species isolated in this study were previously isolated from these four sites (47/198,

[revised manuscript text omitted]

relatively accurate, causing our comparison between the two methods with only genus
information, giving up the species information from culturomics method.
Metagenomic sequencing combined with culturomics should be preferable for the
parallel comparison at the species level. Moreover, the causal relationship between the
screened microorganisms and the occurrence and development of lung cancer remains
uncertain, and follow-up studies are also required.

In conclusion, we examined the pneumonic and oral microbiota of lung cancer
patients using culturomics and 16S rDNA sequencing. We found alterations in the
microbial community of patients with lung cancer, whose diversity might be site and
pathology dependent. Using culturomics, we found that *Streptococcus* and *Veillonella*
were highly dominant in both pneumonic and oral samples of patients with lung
cancer, which suggests the possible deleterious effects of airway microbial dysbiosis
originating from the oral cavity. We showed that *Prevotella oralis* was isolated only
from the H group and that *Gemella sanguinis* was isolated only from the NSCLC
group, which was consistent with the findings of 16S rDNA sequencing. This study
provides basic data on the microbiota diversity in pneumonic and oral samples from
patients with lung cancer. These might serve as potential bacterial biomarkers and
new targets for lung cancer diagnosis and treatment, and the causative relationships
can be explored using these isolated strains.

**AUTHOR CONTRIBUTIONS**

**Ethics approval and consent to participate**

All procedures performed in studies involving human participants or animals were
approved by the Institutional Review Board of the Peking University School of
Oncology (No. 2018KT89). All patients gave their written consent before inclusion in
the study.

**Consent for publication**

Not applicable.

**Data availability**

All the data generated or analyzed in this study are included in this published article
(or its Supplementary Information files). The read sequences obtained from Illumina
NovaSeq were submitted to the NCBI Sequence Read Archive (SRA) under accession
number PRJNA904049 (BioProject ID)
(<http://www.ncbi.nlm.nih.gov/bioproject/?term=PRJNA904049>).

**Competing Interest**

The authors declare no competing interests.

**Funding**

This research was supported by the National Key Research and Development
Program of China (2021YFC2301000), and the National Natural Science Foundation
of China (31970863 and 81790632).

**Authors' contributions**

Y.J.B. designed research and project outline. R.F.Y and Z.P.W. directed the research.
Y.F.S, J.J.L., Y.J.L., A.T.T, Z.M.L, M.M.L, J.B, and Z.H.W. performed isolation,
deposition and identification. Y.F.T. and Z.Y.P. performed genome analysis. Y.F.S.
and Y.J.B. drafted the manuscript. All authors read and approved the final manuscript.

**Acknowledgements**

We gratefully acknowledge all of the participants and the medical staff who helped
collect the samples in this study.

Reference

- 1. Murray, C.J. and A.D. Lopez, *Alternative projections of mortality and disability by*
*cause 1990-2020: Global Burden of Disease Study*. Lancet (London, England), 1997.
**349**(9064): p. 1498-1504.
- 2. Saus, E., et al., *Microbiome and colorectal cancer: Roles in carcinogenesis and*
*clinical potential*. Molecular Aspects of Medicine, 2019. **69**.
- 3. Hanahan, D., *Hallmarks of Cancer: New Dimensions*. Cancer Discovery, 2022. **12**(1):
p. 31-46.
- 4. Dong, Q., et al., *Host-Microbiome Interaction in Lung Cancer*. Frontiers In
Immunology, 2021. **12**: p. 679829.
- 5. Zheng, L., et al., *Lung microbiome alterations in NSCLC patients*. Scientific Reports,
2021. **11**(1): p. 11736.
- 6. Lee, S.H., et al., *Characterization of microbiome in bronchoalveolar lavage fluid of*
*patients with lung cancer comparing with benign mass like lesions*. Lung Cancer
(Amsterdam, Netherlands), 2016. **102**: p. 89-95.
- 7. Liu, Y., et al., *Lung tissue microbial profile in lung cancer is distinct from emphysema*.
American Journal of Cancer Research, 2018. **8**(9): p. 1775-1787.
- 8. Liu, H.-X., et al., *Difference of lower airway microbiome in bilateral protected*
*specimen brush between lung cancer patients with unilateral lobar masses and*
*control subjects*. International Journal of Cancer, 2018. **142**(4): p. 769-778.
- 9. Huang, D., et al., *The characterization of lung microbiome in lung cancer patients with*
*different clinicopathology*. Am J Cancer Res, 2019. **9**(9): p. 2047-2063.

- 10. Gomes, S., et al., *Profiling of lung microbiota discloses differences in*
*adenocarcinoma and squamous cell carcinoma*. Scientific Reports, 2019. **9**(1): p.
12838.
- 11. Christiani, D.C., *The oral microbiome and lung cancer risk*. Thorax, 2021. **76**(3): p.
216-217.
- 12. Tsay, J.-C.J., et al., *Lower Airway Dysbiosis Affects Lung Cancer Progression*.
Cancer Discovery, 2021. **11**(2): p. 293-307.
- 13. Yang, J., et al., *Dysbiosis of the Salivary Microbiome Is Associated With Non-*
*smoking Female Lung Cancer and Correlated With Immunocytochemistry Markers*.
Frontiers In Oncology, 2018. **8**: p. 520.
- 14. Bilen, M., et al., *The contribution of culturomics to the repertoire of isolated human*
*bacterial and archaeal species*. Microbiome, 2018. **6**(1): p. 94.
- 15. Lagier, J.-C., et al., *Culturing the human microbiota and culturomics*. Nature Reviews.
Microbiology, 2018. **16**: p. 540-550.
- 16. Zhuo, M., et al., *Characterization of Microbiota in Cancerous Lung and the*
*Contralateral Non-Cancerous Lung Within Lung Cancer Patients*. Frontiers In
Oncology, 2020. **10**: p. 1584.
- 17. Chang, Y., et al., *Optimization of Culturomics Strategy in Human Fecal Samples*.
Frontiers In Microbiology, 2019. **10**: p. 2891.
- 18. Chun, J., et al., *Proposed minimal standards for the use of genome data for the*
*taxonomy of prokaryotes*. International Journal of Systematic and Evolutionary
Microbiology, 2018. **68**(1): p. 461-466.

- 19. Fonkou, M.D., et al., *Repertoire of bacterial species cultured from the human oral*
*cavity and respiratory tract*. Future Microbiology, 2018. **13**: p. 1611-1624.
- 20. Lagier, J.-C., et al., *Culture of previously uncultured members of the human gut*
*microbiota by culturomics*. Nature Microbiology, 2016. **1**: p. 16203.
- 21. Dubourg, G., et al., *Deciphering the Urinary Microbiota Repertoire by Culturomics*
*Reveals Mostly Anaerobic Bacteria From the Gut*. Frontiers In Microbiology, 2020. **11**:
p. 513305.
- 22. Morand, A., et al., *Human Bacterial Repertoire of the Urinary Tract: a Potential*
*Paradigm Shift*. Journal of Clinical Microbiology, 2019. **57**(3).
- 23. Price, T.K., et al., *The Clinical Urine Culture: Enhanced Techniques Improve*
*Detection of Clinically Relevant Microorganisms*. Journal of Clinical Microbiology,
2016. **54**(5): p. 1216-1222.
- 24. Kitagawa, K., et al., *Improved bacterial identification directly from urine samples with*
*matrix-assisted laser desorption/ionization time-of-flight mass spectrometry*. Journal
of Clinical Laboratory Analysis, 2018. **32**(3).
- 25. Goubet, A.-G., et al., *The impact of the intestinal microbiota in therapeutic responses*
*against cancer*. Comptes Rendus Biologies, 2018. **341**(5): p. 284-289.
- 26. Georgiou, K., et al., *Gut Microbiota in Lung Cancer: Where Do We Stand?*
International Journal of Molecular Sciences, 2021. **22**(19).
- 27. Pizzo, F., et al., *Role of the Microbiota in Lung Cancer: Insights on Prevention and*
*Treatment*. International Journal of Molecular Sciences, 2022. **23**(11).
- 28. Goto, T., *Microbiota and lung cancer*. Seminars In Cancer Biology, 2022. **86**(Pt 3).

- 29. Mortensen, M.S., et al., *The developing hypopharyngeal microbiota in early life.*
Microbiome, 2016. **4**(1): p. 70.
- 30. Venkataraman, A., et al., *Application of a neutral community model to assess*
*structuring of the human lung microbiome.* MBio, 2015. **6**(1).
- 31. Zhang, J., et al., *Differential Oral Microbial Input Determines Two Microbiota*
*Pneumo-Types Associated with Health Status.* Advanced Science (Weinheim,
Baden-Wurttemberg, Germany), 2022. **9**(32): p. e2203115.
- 32. Segal, L.N., et al., *Enrichment of the lung microbiome with oral taxa is associated*
*with lung inflammation of a Th17 phenotype.* Nature Microbiology, 2016. **1**: p. 16031.
- 33. Ramírez-Labrada, A.G., et al., *The Influence of Lung Microbiota on Lung*
*Carcinogenesis, Immunity, and Immunotherapy.* Trends In Cancer, 2020. **6**(2): p. 86-
97.
- 34. Dickson, R.P., et al., *The Microbiome and the Respiratory Tract.* Annual Review of
Physiology, 2016. **78**: p. 481-504.
- 35. Bellali, S., et al., *Running after ghosts: are dead bacteria the dark matter of the*
*human gut microbiota?* Gut Microbes, 2021. **13**(1).
- 36. Xi Zhen, L., et al., *Symphysis Pubis Diastasis Due to Parvimonas micra Infection; an*
*Unusual Suspect.* Journal of Clinical Rheumatology : Practical Reports On Rheumatic
& Musculoskeletal Diseases, 2021. **27**(3): p. e98-e99.
- 37. Yu, Q., et al., *Severe pneumonia caused by Parvimonas micra: a case report.* BMC
Infectious Diseases, 2021. **21**(1): p. 364.
- 38. Zhao, L., et al., *Parvimonas micra promotes colorectal tumorigenesis and is*

- *associated with prognosis of colorectal cancer patients*. *Oncogene*, 2022. **41**(36): p.
4200-4210.
- 39. Löwenmark, T., et al., *Parvimonas micra as a putative non-invasive faecal biomarker*
*for colorectal cancer*. *Scientific Reports*, 2020. **10**(1): p. 15250.
- 40. Chang, Y., et al., *Parvimonas micra activates the Ras/ERK/c-Fos pathway by*
*upregulating miR-218-5p to promote colorectal cancer progression*. *Journal of*
*Experimental & Clinical Cancer Research : CR*, 2023. **42**(1): p. 13.
- 41. Tsay, J.-C.J., et al., *Airway Microbiota Is Associated with Upregulation of the PI3K*
*Pathway in Lung Cancer*. *American Journal of Respiratory and Critical Care Medicine*,
2018. **198**(9): p. 1188-1198.
- 42. Jurado-Martín, I., M. Sainz-Mejías, and S. McClean, : *An Audacious Pathogen with*
*an Adaptable Arsenal of Virulence Factors*. *International Journal of Molecular*
*Sciences*, 2021. **22**(6).
- 43. Biswas, L. and F. Götz, *Molecular Mechanisms of and Interactions in Cystic Fibrosis*.
*Frontiers In Cellular and Infection Microbiology*, 2021. **11**: p. 824042.
- 44. Das, S., et al., *A prevalent and culturable microbiota links ecological balance to*
*clinical stability of the human lung after transplantation*. *Nature Communications*,
2021. **12**(1): p. 2126.
- 45. Reynolds, D. and M. Kollef, *The Epidemiology and Pathogenesis and Treatment of*
*Pseudomonas aeruginosa Infections: An Update*. *Drugs*, 2021. **81**(18): p. 2117-2131.
- 46. Curran, C.S., T. Bolig, and P. Torabi-Parizi, *Mechanisms and Targeted Therapies for*
*Pseudomonas aeruginosa Lung Infection*. *American Journal of Respiratory and*

- Critical Care Medicine, 2018. **197**(6): p. 708-727.
- 47. Lu, H., et al., *Alterations of the Human Lung and Gut Microbiomes in Non-Small Cell*
*Lung Carcinomas and Distant Metastasis*. Microbiology Spectrum, 2021. **9**(3): p.
e0080221.
- 48. Ferrero, M.C., et al., *Pathogenesis and immune response in Brucella infection*
*acquired by the respiratory route*. Microbes and Infection, 2020. **22**(9): p. 407-415.
- 49. Roop, R.M., et al., *Uncovering the Hidden Credentials of Brucella Virulence*.
Microbiology and Molecular Biology Reviews : MMBR, 2021. **85**(1).
- 50. Maraki, S., et al., *Bicuspid aortic valve endocarditis caused by Gemella sanguinis:*
*Case report and literature review*. Journal of Infection and Public Health, 2019. **12**(3):
p. 304-308.
- 51. Antunes, H.S., et al., *Total and Specific Bacterial Levels in the Apical Root Canal*
*System of Teeth with Post-treatment Apical Periodontitis*. Journal of Endodontics,
2015. **41**(7): p. 1037-1042.
- 52. Issa, E., T. Salloum, and S. Tokajian, *From Normal Flora to Brain Abscesses: A*
*Review of*. Frontiers In Microbiology, 2020. **11**: p. 826.
- 53. Yadava, K., et al., *Microbiota Promotes Chronic Pulmonary Inflammation by*
*Enhancing IL-17A and Autoantibodies*. American Journal of Respiratory and Critical
Care Medicine, 2016. **193**(9): p. 975-987.
- 54. Jin, C., et al., *Commensal Microbiota Promote Lung Cancer Development via $\gamma\delta$ T*
*Cells*. Cell, 2019. **176**(5).

**Fig 1.** Summary of culturomics methods and workflow.

**Fig 2.** Bacteria identified from the BALF samples. **A)** Phylogenetic tree and **B)**
proportion of 198 bacterial species isolated from the BALF samples listed according
to their phylum. **C)** UpSet plot showing the shared cultured species among human
oral/respiratory, gut, urine, and vagina.

**Fig 3.** Variations in microbial composition in different anatomical sites. **A)**
Taxonomic composition at the phylum level in BALF and oral samples based on 16S
rDNA amplicon sequencing (Top 30). **B)** The Shannon diversity index of BALF and
oral samples. p values were calculated using the Wilcoxon test. $*p < 0.05$. **C)** PCoA
analysis of samples from three different anatomical site in patients with lung cancer.
ANOSIM was performed to statistically evaluate significant difference. **D)** Notched
box plots illustrating the differences of the significantly 2 genus relative abundances
in three different anatomy sites. $*p < 0.05$. **E)** Proportions of bacterial species
isolated from C, H, and O samples listed according to their phylum. **F)** Venn diagram
of the culturable bacterial species from the three different anatomical sites.
Abbreviations: C, H, O: samples from the cancerous site and contralateral healthy
controls from lungs and the oral site, respectively, of patients with lung cancer.

**Fig 4.** Differentially abundant taxonomy between cancer and paired healthy lung. **A)**
LEfSe was used to identify the bacterial microbiota that significantly differed between
the cancerous and healthy lung. Only taxa meeting a significant LDA threshold value
of >3.5 and $p < 0.05$ are shown. **B)** Heatmap analysis of lung microbiota at C and H
sites of the patients with lung cancer based on culturomics ($n = 15$). Each row
represents an individual species with the higher isolated rate in the two groups. Top 25
bacterial species were recovered from either group. Transition from blue to red
represents the frequency of culturomics recovery from 0 to 1. Statistical comparisons
were made using the Fischer's exact test with FDR correction. $*p < 0.05$

**Fig 5.** Characterization of lung microbiota in different lung cancer subtypes. **A)**
Taxonomic composition at the genera level in NSCLC and SCLC lung samples. **B)**
Venn diagram of the culturable bacterial species isolated only from NSCLC and only
from SCLC lung samples. **C)** Differences in the relative abundance of the genera
corresponding to the species isolated from NSCLC and SCLC samples. Differential
microbial taxa were identified using paired t-test, and the p -values were adjusted for
multiple comparison using the FDR. **D)** Taxonomic composition at genera level in
ADC and SCC lung samples. **E)** The LEfSe was used to identify the bacterial
microbiota that significantly differed between NSCLC and SCLC. Only taxa meeting
a significant LDA threshold value of >4 and $p < 0.05$ are shown.

**Table 1.** Clinical characteristics of patients.

Variable	NSCLC	SCLC	Non
N	16	7	2
Age-mean (SD)	65 (7.0)	67 (12.8)	69 (8.2)
Gender			
Male, n (%)	9 (56%)	5 (72%)	2
Female, n (%)	7 (44%)	2 (14%)	0
Smoking			
Current or former Smoker, n (%)	12(75%)	6(86%)	1
Never smoker, n (%)	6(25%)	1(14%)	1
Pathological diagnosis			
Adenocarcinoma, n (%)	8(50%)	—	—
Squamous cell carcinoma, n (%)	7(44%)	—	—
Unidentified	1(6%)	—	—
Distant			
MO	4(25%)	2(29%)	—
M1	12(75%)	5(71%)	—

**Table 2.** Summary of the proportion of the TOP20 bacterium in BALF samples and
 the related proportion in oral cavity.

Genera	Species	C	H	O
	Streptococcus salivarius	60.00%	60.00%	73.33%
	Streptococcus pseudopneumoniae	53.33%	40.00%	33.33%
	Streptococcus parasanguinis	93.33%	86.67%	100.00%
Streptococcus	Streptococcus oralis	93.33%	93.33%	100.00%
	Streptococcus mitis	93.33%	86.67%	100.00%
	Streptococcus gordonii	53.33%	53.33%	80.00%
	Streptococcus constellatus	66.67%	73.33%	93.33%
	Streptococcus anginosus	66.67%	66.67%	100.00%
Veillonella	Veillonella parvula	60.00%	60.00%	73.33%
	Veillonella atypica	66.67%	80.00%	66.67%
Solobacterium	Solobacterium moorei	66.67%	60.00%	73.33%
Slackia	Slackia exigua	60.00%	73.33%	60.00%
Parvimonas	Parvimonas micra	80.00%	80.00%	93.33%
Mogibacterium	Mogibacterium diversum	53.33%	46.67%	53.33%
Granulicatella	Granulicatella adiacens	66.67%	53.33%	46.67%
Gemella	Gemella morbillorum	66.67%	53.33%	66.67%
Fusobacterium	Fusobacterium nucleatum	40.00%	66.67%	80.00%
Dialister	Dialister invisus	73.33%	53.33%	73.33%
Anaeroglobus	Anaeroglobus geminatus	53.33%	60.00%	73.33%
Actinomyces	Actinomyces odontolyticus	80.00%	80.00%	66.67%

Abbreviations: C, H, O: samples from the cancerous site and contralateral healthy
 controls from lungs and the oral site, respectively, of patients with lung cancer.

629